# FINDER: FEATURE INFERENCE ON NOISY DATASETS USING EIGENSPACE RESIDUALS

## ABSTRACT

"Noisy" datasets (regimes with low signal to noise ratios, small sample sizes, faulty data collection, etc) remain a key research frontier for classification methods with both theoretical and practical implications. We introduce FINDER, a rigorous framework for analyzing generic classification problems, with tailored algorithms for noisy datasets. FINDER incorporates fundamental stochastic analysis ideas into the feature learning and inference stages to optimally account for the randomness inherent to all empirical datasets. We construct "stochastic features" by first viewing empirical datasets as realizations from an underlying random field (without assumptions on its exact distribution) and then mapping them to appropriate Hilbert spaces. The Karhunen-Loève (KL) transform breaks these stochastic features into computable irreducible components, which allow classification over noisy datasets via an eigen-decomposition: data from different classes resides in distinct regions, identified by analyzing the spectrum of the associated operators. We validate FINDER on several challenging, data-deficient scientific domains, producing state of the art breakthroughs in: (i) Alzheimer's Disease stage classification, (ii) Remote sensing detection of deforestation. We end with a discussion on when FINDER is expected to outperform existing methods, its failure modes, and other limitations.

## 1 INTRODUCTION

Classification problems are of significant interest across a variety of scientific and commercial fields, especially when concerned with "noisy" datasets: settings where the nominal data dimension $F \gg$ than the sample size $N$, datasets with poor signal to noise ratios, etc. Deep/Machine Learning (ML) methods are particularly known to be susceptible in data-deficient settings LeCun et al. (2014); Ng (2004) and thus techniques for performance improvements in these regimes remain of strong interest.

We present a multi-faceted novel development within these contexts: a generic and versatile theory for discussing classification problems with applications in treating noisy datasets, validated over a collection of challenging and significant scientific datasets. Broadly speaking, we blend standard feature inference/construction methods with the Kosambi-Karhunen-Loève theorem Loève (1978) from stochastic analysis, by rigorously defining and building "stochastic features" that can help classify the underlying structures while seeing through the inherent "blurriness" of noisy datasets. We begin with binary classification since many classification problems reduce to a set of binary ones.

Binary classification involves classifying an input object into one of two classes, $\{\mathbf{A}, \mathbf{B}\}$, $\{0, 1\}$, etc, based on a list of numeric quantities associated with that object. For large $F$, this becomes computationally intractable as $N$ may be too limited for machine training. Principal component analysis (PCA) was developed largely as a way to effectively reduce the nominal dimension of a given dataset with minimal information loss Kokoszka and Reimherr (2017). Further developments involved viewing data not as a vector in $\mathbb{R}^F$, but as a function from a closed interval $[a, b]$ to $\mathbb{R}$, sampled at $F$ points. Such methods for analyzing, constructing machines from, and making predictions based on data comprise the field of functional data analysis (FDA) Kokoszka and Reimherr (2017); Horváth and Kokoszka (2012). FDA is often preferred when $F \gg N$: even linear discriminant analysis can be outperformed by the much simpler naive Bayesian classification Bickel and Levina (2004).

The Kosambi-Karhunen-Loève expansion (KLE) Schwab and Todor (2006) is a fundamental result in FDA, mildly generalized by us as Thm. 2.1. It implies that our stochastic features admit a Fourier series like expansion in terms of simpler, computable elements. We pair Thm. 2.1 with novel

algorithms for constructing/finding stochastic features with in-built class separability (to the extent such underlying features can exist for the available data). We further prove that digitization does not limit the usefulness of these results and show how optimal choices for truncation may be made.

We test our approach on noisy datasets of significant scientific interest, with major improvements on existing state of the art results. FINDER is generic, but geared towards such data-deficient or otherwise noisy settings, its relative performance advantages expectedly increasing with the "noise".

Another advantage lies in the efficiencies it may unlock for otherwise computationally intensive tasks. The inherent robustness to noise and in-built class separability implies nominally intractable datasets can become amenable even to fundamental ML algorithms like support vector machines (SVM) or hidden Markov models (HMM), which can then be used with dramatic improvements in accuracy. Hence, while FINDER comes with additional construction costs of its own, the fact that it can be packaged with simpler methods like SVM provides a pathway to manageable computational costs.

FINDER is also relatively robust to unbalanced data: it is a functional analytic schema and the number of available samples per class is a matter of concern only insomuch that the "noisiness" of the class changes with that number. Numerical experiments validating our claims are provided in Sec. 3.

## 2 A Mathematical Framework for Feature Inference

FINDER is a 3-step framework: 1) Dataset acquisition, 2) Feature construction, 3) Classification.

We begin by assuming that the dataset $\mathcal{D}$, a subset of some nominal space $U$, is a random realization of a complete probability space $(\Omega, \mathcal{F}, \mathbb{P})$, without any assumptions on the underlying data distribution. $U$ is usually a Euclidean space, but could be a manifold, spatio-temporal domain, etc. Formally, we view this realization of the random field through a map $v_1 : \Omega \to U$. We then use another map $v_2$ to map $\mathcal{D}$ to some apt Hilbert space $\mathcal{H}$, such that different classes get mapped to disjoint regions in $\mathcal{H}$. Readers may recognize this as a standard feature construction task or kernel trick. The composition $v := v_1 \circ v_2$ is called a ***stochastic feature*** (and is Bochner measurable[A.1]), if $v \in L^2(\Omega, \mathcal{H})$.

Classifiers are then simply maps from $\mathcal{H}$ to $\{0, 1\}$: usually via a machine from $\mathcal{H}$ to $[0, 1]$, with a separatrix $t \in (0, 1)$. Figure 1 summarizes FINDER as a whole, but good features make Step 3 trivial, so our focus is on the composed Steps 1 and 2: the creation and computability of stochastic features.

The novelty of our work is in that our features directly incorporate the stochasticity through which $\mathcal{D}$ is generated. Thus, most binary classification problems fall within the ambit of our framework, while it becomes especially well-suited to handling noisy datasets. FINDER is agnostic to the choice of $(\Omega, \mathcal{F}, \mathbb{P})$ if it is a complete probability space. $\mathcal{H} = L^2([a, b])$ or $\mathbb{R}^F$ are usual choices if $F$ is large.

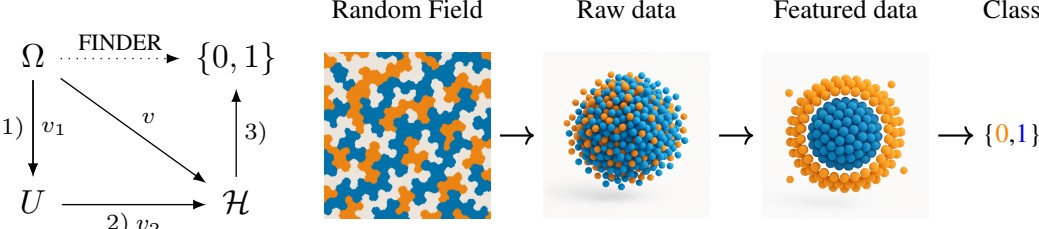

Figure 1: A schematic for FINDER and a visual perspective on classification as a multi-stage process.

Unfortunately, constructing stochastic features $v$ can be prohibitively expensive. However, our goal is classification and the eigen-decomposition of $v$ via the KLE provides an optimally efficient short-cut.

### 2.1 A generalized Kosambi-Karhunen–Loève expansion theorem

We will first need to define the notions of expectation and covariance to formally discuss the KLE:

**Definition 2.1.** *Let $\langle u, v \rangle_{\mathcal{H}}$ and $\int_{\Omega} \langle u, v \rangle_{\mathcal{H}} d\mathbb{P}$ be the inner products on $\mathcal{H}$ and $L^2(\Omega, \mathcal{H})$ respectively. The Expectation Operator for $v$ is the Bochner integral:* $\mathbb{E} : L^2(\Omega, \mathcal{H}) \to \mathcal{H}, \mathbb{E}(v) = \int_{\Omega} v(\omega) d\mathbb{P}$. *The Covariance Operator is given by:* $\quad \mathcal{C}_v : \mathcal{H} \to \mathcal{H}, \quad \mathcal{C}_v(e) = \mathbb{E}(\langle v - \mathbb{E}(v), e \rangle_{\mathcal{H}} (v - \mathbb{E}(v)))$.

For example, if $\mathcal{H} = L^2([a,b])$, $\mathcal{C}_v$ is the kernel operator whose kernel is $\mathcal{K}(x,y) = \mathbb{E}[(v(x,\omega) - \mathbb{E}[v(x,\omega)])(v(y,\omega) - \mathbb{E}[v(y)])]$ and produces a map $\mathcal{C}_v(f)$ s.t. $\mathcal{C}_v(f)(x) := \int_a^b \mathcal{K}(x,y)f(y)\,dy$.

We now state a mildly generalized KLE (App. A), dropping the usual separability assumptions on $\mathcal{H}$.

**Theorem 2.1.** *Let $v \in L^2(\Omega, \mathcal{H})$ be Bochner measurable. Then, there exists $R \in \mathbb{N} \cup \aleph_0$ such that*

$$v(\omega) = \mathbb{E}(v) + \sum_{r=1}^{R} \lambda_r^{1/2} Y_r(\omega) \phi_r, \qquad \lambda_r \in (0, \infty), \quad \sum_r \lambda_r < \infty, \quad \lambda_1 \geq \lambda_2 \geq ... \quad (1)$$

*where $\{Y_r\}_{r=1}^{R}, \{\phi_r\}_{r=1}^{R}$ are orthonormal sets in $L^2(\Omega), \mathcal{H}$ respectively, with $\mathbb{E}[Y_r] = 0$ for all $r$.*

For example, if $\mathcal{H} = L^2([a,b])$, then $v$ is simply a measurable map $v : [a,b] \times \Omega \to \mathbb{R}$, such that

$$v(x,\omega) = \mathbb{E}(v) + \sum_{r=1}^{R} \lambda_r^{1/2} Y_r(\omega) \phi_r(x), \qquad Y_r \in L^2(\Omega),\ \phi_r \in L^2([a,b])$$

Thm. 2.1 justifies $L^2(\Omega, \mathcal{H})$ as the setting to source stochastic features from. We will need some intermediate results to better understand its proof and the computability of KLE. We begin by showing the isomorphism of $L^2(\Omega, \mathcal{H})$ to the space of Hilbert-Schmidt operators $\mathrm{HS}(\mathcal{H}, L^2(\Omega))$ in App. A.2:

**Lemma 2.1.** *Let $(\Omega, \mathcal{F}, \mathbb{P})$ be a complete probability space and $\mathcal{H}$ be a Hilbert space. Then, $L^2(\Omega, \mathcal{H})$ is isometrically isomorphic to $\mathrm{HS}(\mathcal{H}, L^2(\Omega))$.*

Lemma 2.1 implies there is an Hilbert-Schmidt operator $H_v \in \mathrm{HS}(\mathcal{H}, L^2(\Omega))$ in correspondence with every $v$. $H_v$ is Hilbert-Schmidt means it is compact: the spectral theorem then generates our KLE directly through a generalized Singular Value Decomposition (SVD), discussed in App. A.3.

Let $H_v$ be the map $e \mapsto \langle v - \mathbb{E}(v), e \rangle_{\mathcal{H}}$. We promptly see that $\mathcal{C}_v = H_v^* H_v$. Thus, $\lambda_r^{1/2}, \phi_r$ are just the singular values and right singular vectors of $H_v$ (App. A.4), while $Y_r$ are the left singular vectors.

However, we have established properties over infinite dimensional spaces that no computer can directly make use of. Fortunately, these properties pass over to the truncations we will necessarily make: instead of having to build a possibly infinite dimensional Hilbert space $\mathcal{H}$ and projecting onto a subspace $\mathcal{H}_M$, we can work directly on our chosen $\mathcal{H}_M$ (see Sec. A.5). Let $P_{\mathcal{S}}$ represent projection onto some subspace $\mathcal{S} \subset \mathcal{H}$. Lemma 2.2 tells us the optimal $M$ dimensional subspace to work in:

**Lemma 2.2.** *Let $v \in L^2(\Omega, \mathcal{H})$ and $\mathcal{S} \subset \mathcal{H}$ be an arbitrary $M$ dimensional subspaces. Then $\mathcal{S}^* = \mathrm{Span}\{\phi_r\}_{r=1}^{M} = \underset{\mathcal{S}}{\mathrm{argmin}} \|v - P_{\mathcal{S}}v\|_{L^2(\Omega, \mathcal{H})}$.*

We are now ready to present the class separation identities that turn these results into applications.

## 2.2 CLASSIFICATION AS FEATURED SEPARATION

We begin by viewing all class $\mathbf{A}$ elements in $\mathcal{H}$ as images of some stochastic feature $v^{\mathbf{A}}$ and class $\mathbf{B}$ elements as images of some $v^{\mathbf{B}}$. FINDER centers the dataset on class $\mathbf{A}$ by setting $v^{\mathbf{A}} \to v^{\mathbf{A}} - \mathbb{E}(v^{\mathbf{A}})$ and $v^{\mathbf{B}} \to v^{\mathbf{B}} - \mathbb{E}(v^{\mathbf{A}})$, using the training data to do so. The inherent assumption here is that the training set data allows an adequate estimation of $\mathbb{E}(v^{\mathbf{A}})$. Immediately:

$$v^{\mathbf{A}} = \sum_{r=1}^{R_{\mathbf{A}}} \lambda_r^{\mathbf{A}\,1/2} Y_r^{\mathbf{A}} \phi_r^{\mathbf{A}}, \qquad v^{\mathbf{B}} = \mathbb{E}(v^{\mathbf{B}}) + \sum_{r=1}^{R_{\mathbf{B}}} \lambda_r^{\mathbf{B}\,1/2} Y_r^{\mathbf{B}} \phi_r^{\mathbf{B}} \qquad (2)$$

Let $\mathcal{H}_{\mathbf{A}} = \mathrm{Span}\{\phi_r^{\mathbf{A}}\}_{r=1}^{M_{\mathbf{A}}}$. In practice, we choose some finite $M_{\mathbf{A}} \in \mathbb{N}$ by truncating the KLE (Lemma 2.2), based on the acceptable error tolerance for the problem at hand. Our goal is to construct a residual subspace $\mathcal{H}_{\mathrm{res}} \subset \mathcal{H}_{\mathbf{A}}^{\perp}$ such that class $\mathbf{B}$ elements present a different profile in $\mathcal{H}_{\mathrm{res}}$ compared to class $\mathbf{A}$: the usual choice is to have $v_{\mathbf{B}}$ concentrate in $\mathcal{H}_{\mathrm{res}}$ and/or have a different spectral profile over $\mathcal{H}_{\mathrm{res}}$ than $v^{\mathbf{A}}$ (for example, in Figure 1, both classes share mean, but not distribution).

$\mathcal{H}_{\text{res}}$ is generated from $\mathcal{H}_{\mathbf{A}}^{\perp}$ with some truncated subspace dimension $M_{\text{res}}$. Intuitively, $\mathcal{H}_{\text{res}}$ represents the portion of $\text{Span}\left\{\phi_r^{\mathbf{B}}\right\}$ that is **not** overlapping with $\text{Span}\left\{\phi_r^{\mathbf{A}}\right\}$: in short, portions that make class $\mathbf{A}, \mathbf{B}$ "different" in some distributional sense (otherwise they are indistinguishable in $\mathcal{H}$ anyway).

Thus, we need a formal result/algorithm that shows stochastic features that map distinct classes to distinct regions in $\mathcal{H}$ have spectral profiles concentrating with computable differences within some probability. We invoke Markov's inequality to establish the following Lemma (App. B.1):

**Lemma 2.3.** *Let $\mathcal{S} \subset \mathcal{H}$ be a finite-dimensional subspace with orthonormal basis $\{s_m\}_{m=1}^{M_{res}}$. Then*

$$
\begin{aligned}
\Pr(\|P_{\mathcal{S}}(v^{\mathbf{A}} - \mathbb{E}(v^{\mathbf{A}}))\|_{\mathcal{H}}^2 \geq \varepsilon^2) \quad &\leq \varepsilon^{-2} \sum_{m=1}^{M_{res}} \sum_{r=1}^{R_{\mathbf{A}}} \lambda_r^{\mathbf{A}} \left\langle \phi_r^{\mathbf{A}}, s_m \right\rangle_{\mathcal{H}}^2 \\
\Pr(\|P_{\mathcal{S}}(v^{\mathbf{B}} - \mathbb{E}(v^{\mathbf{B}}))\|_{\mathcal{H}}^2 \geq \varepsilon^2) \quad &\leq \varepsilon^{-2} \sum_{m=1}^{M_{res}} \sum_{r=1}^{R_{\mathbf{B}}} \lambda_r^{\mathbf{B}} \left\langle \phi_r^{\mathbf{B}}, s_m \right\rangle_{\mathcal{H}}^2
\end{aligned}
\tag{3}
$$

*for any $\varepsilon > 0$, where $\left\{\lambda_r^{\mathbf{A}}, \phi_r^{\mathbf{A}}\right\}_{r=1}^{R_{\mathbf{A}}}, \left\{\lambda_r^{\mathbf{B}}, \phi_r^{\mathbf{B}}\right\}_{r=1}^{R_{\mathbf{B}}}$ are the eigen-pairs of $\mathcal{C}_{v^{\mathbf{A}}}$ and $\mathcal{C}_{v^{\mathbf{B}}}$ respectively.*

Lemma 2.3 suggests that if both RHS of (3) are small for $\mathcal{S} \subseteq \mathcal{H}$, then mapped stochastic features (i.e. $P_{\mathcal{S}} v^{\mathbf{A}}$ or $P_{\mathcal{S}} v^{\mathbf{B}}$) concentrate around their respective mapped class expectation with high probability.

**Remark 2.1.** *The value of $\lambda_r$ is simply the variance of the real-valued random variable $\langle v, \phi_r \rangle_{\mathcal{H}}$. Because the KLE acts as a generalized SVD, the eigen-pairs $(\lambda_r, \phi_r)$ capture the subspace in which $v$ tends to concentrate in, along with the spread of $v$ within that subspace. This can be used, as in the Markov bound (3) above, to place deterministic bounds on the probability that $v$ lies in certain regions of $\mathcal{H}$ without knowing the underlying distribution of $v$. For many datasets, only the first few eigenvalues $\lambda_r^{\mathbf{A}}$ are non-negligible. Thus, if $\{s_m\}_{m=1}^{M_{res}}$ is an orthonormal set in $\mathcal{H}_{\mathbf{A}}^{\perp}$, then $\sum_{m=1}^{M_{res}} \sum_{r=1}^{R_{\mathbf{A}}} \lambda_r^{\mathbf{A}} \left\langle \phi_r^{\mathbf{A}}, s_m \right\rangle^2 \leq \sum_{r > M_{\mathbf{A}}} \lambda_r^{\mathbf{A}}$. This is the one of the rationales behind the three FINDER variants we will present later.*

The Markov bounds hold regardless of the distribution of the stochastic components (the sequence $Y_r$ in the KLE) of $v$. $Y_r$ having 0 expectation essentially filters them out of the inequality. This distribution-agnostic aspect of FINDER serves to reduce computational complexity by eliminating the need to estimate the $Y_r$, a process which can be expensive in the absence of sufficient data and one that may need additional, potentially unrealistic, assumptions on the probability space $(\Omega, \mathcal{F}, \mathbb{P})$.

However, the saved costs come with a large disadvantage: 3 is rarely a tight bound. So while we may not need assumptions on $(\Omega, \mathcal{F}, \mathbb{P})$, better understanding of $Y_r$ can get more useful bounds than (3).

## 2.3 CONSTRUCTING RESIDUAL EIGENSPACES AND IMPLEMENTATIONS

FINDER comes with inherent flexibility of implementation since we may tile $\mathcal{H}_{\text{res}}$ in a variety of ways (we may even nonlinearly parameterize $\mathcal{H}_{\text{res}}$ using deep neural networks). However, in this work we use the following linearly parameterized approaches (complexity analysis in Sec. B.4):

- Direct Residual Subspaces: Initially proposed in Lakhina et al. (2004) in PCA contexts. We simply extend it to the KLE setting by taking $\mathcal{H}_{\text{res}} = \mathcal{H}_{\mathbf{A}}^{\perp} = (\text{Span}\left\{\phi_r^{\mathbf{A}}\right\}_{r=1}^{M_{\mathbf{A}}})^{\perp}$.
- Multi-Level Subspaces (MLS): This approach adapts the algorithm in Tausch and White (2003) for constructing a basis for the residual subspace. Class $\mathbf{A}$ and $\mathbf{B}$ are then projected onto this subspace and used to train the classifier. It is detailed in Sec. B.2.
- Anomalous Class Adapted (ACA-S and ACA-L): Results in Sec. 2.1, 3 lead to two novel, related methods, each relying on two successive projections to class $\mathbf{A}$ and $\mathbf{B}$ samples, as described in Alg. 1 and detailed in Section B.3.

FINDER variants are tested by training on both unbalanced and balanced datasets (SVMs will be our usual choice for conventional feature construction, but we will occasionally leverage HMMs too). For Balanced, $N_{\mathbf{A}} - N_{\mathbf{B}} - 1$ samples of Class $\mathbf{A}$ are used to estimate $\mathcal{C}_{v^{\mathbf{A}}}$ and the remaining

---

**Algorithm 1** ACA Algorithm for Residual Eigenspace Construction

---

**Inputs:** $M_{\mathbf{A}} \in \{1, \ldots, \dim(\mathcal{H})\}$, $M_{\text{res}} \in \{1, \ldots, \dim(\mathcal{H}) - M_{\mathbf{A}}\}$
$v_i \leftarrow P_{\mathcal{H}_{\mathbf{A}}^{\perp}} v_i$
**if** ACA-S **then**
$\quad \mathcal{H}_{\text{res}} = \operatorname{argmin} \left\{ \|P_{\mathcal{S}}(v^{\mathbf{B}} - \mathbb{E}(v^{\mathbf{B}}))\|_{L^2(\Omega, \mathcal{H})}^2 \right\}$ s.t. $\mathcal{S} \leq \mathcal{H}_{\mathbf{A}}^{\perp}, \ \dim(\mathcal{S}) = M_{\text{res}}$
**else if** ACA-L **then**
$\quad \mathcal{H}_{\text{res}} = \operatorname{argmax} \left\{ \|P_{\mathcal{S}}(v^{\mathbf{B}} - \mathbb{E}(v^{\mathbf{B}}))\|_{L^2(\Omega, \mathcal{H})}^2 \right\}$ s.t. $\mathcal{S} \leq \mathcal{H}_{\mathbf{A}}^{\perp}, \ \dim(\mathcal{S}) = M_{\text{res}}$
**end if**
$v_i \leftarrow P_{\mathcal{H}_{\text{res}}} v_i$

---

$$\boxed{\begin{array}{c}\text{Raw features: } v^{\mathbf{A}}, v^{\mathbf{B}} \\ \text{Choose } M_{\mathbf{A}}\end{array}} \xrightarrow{v \to P_{\mathcal{H}_{\mathbf{A}}^{\perp}} v} \boxed{\text{Choose } M_{\text{res}}} \xrightarrow{v \to P_{\mathcal{H}_{\text{res}}} v} \boxed{\begin{array}{c}\text{Transformed} \\ \text{Features}\end{array}}$$

---

$N_{\mathbf{B}} - 1$ Class $\mathbf{A}$ samples and $N_{\mathbf{B}} - 1$ Class $\mathbf{B}$ samples are used to train the SVM. For Unbalanced, all $N_{\mathbf{A}} - 1$ Class $\mathbf{A}$ samples are used to estimate $\mathcal{C}_{v^{\mathbf{A}}}$ and all $N_{\mathbf{A}} - 1$ Class $\mathbf{A}$ samples and all $N_{\mathbf{B}} - 1$ Class $\mathbf{B}$ samples are used to train the SVM. In both the Balanced and Unbalanced regime, $\mathcal{C}_{v^{\mathbf{B}}}$ is estimated using all $N_{\mathbf{B}} - 1$ Class $\mathbf{B}$ samples. See Section C.1 and Figure 4 for a detailed description.

## 3 APPLICATIONS AND NUMERICAL EXPERIMENTS

We exemplify FINDER on several noisy datasets, picked for their scientific significance and noted resistance to a variety of standard classification methods. Performance on each problem will be compared against the current state of the art results. We pair FINDER with simple ML methods, allowing us to test the two-fold claims we made regarding its robustness to noise and its capacity for making complex datasets more amenable to simpler ML methods. To assess classification performance, we performed leave-(one)-pair-out cross-validation (LPOCV) on standardized data.

For comparative purposes, our benchmark methods will usually comprise of 1) a linear SVM, 2) SVM with RBF (SVM Radial), 3) LogitBoost, 4) RUSBoost, and 5) random forest (BAG) trained on raw features. We present and discuss two regimes where FINDER produced significant breakthroughs, while App. D considers its impact and limitations across a wider variety of settings and applications.

### 3.1 AD CLASSIFICATION FROM BLOOD PLASMA PROTEIN DATA

Our first suite of tests employs proteomics data from the Alzheimer's Disease Neuroimaging Initiative (ADNI) Petersen et al. (2010). The features correspond to 146 blood plasma biomarkers. The final cohort distribution includes 54 Cognitive Normal (CN), 96 Alzheimer's Disease (AD), and a mix of 346 Mild Cognitive Impairment (MCI) to Late MCI participants, for simplicity referred to as LMCI.

Early and accurate AD state classification is critical for the tens of millions of people suffering from or at risk of AD, particularly since early detection significantly improves prognosis. High accuracy techniques to distinguish between CN and LMCI with minimally invasive methods like blood tests are a prominent area of research, while tests for CN vs Early MCI would be even more significant.

In our experiments, we have $U = \mathcal{H} = \mathcal{H}_M = \mathbb{R}^{146}$. Furthermore, $\mathcal{H}_{\text{res}} \leq \mathcal{H}_{\mathbf{A}}^{\perp}$, where $\dim(\mathcal{H}_{\mathbf{A}}) = M_{\mathbf{A}} = 5$. Table 1 summarizes the best AUC obtained on the ADNI dataset by various FINDER variants and the best AUC performance by the entire set of benchmark learners.

We note that the AUC results we obtain with ACA and MLS are significantly higher under FINDER than the benchmark approach. At the same time, the results in Table 1 demonstrate that the run time under FINDER also improves significantly in all three cohorts. This efficiency is especially remarkable in the AD vs. CN and AD vs. LMCI cohorts, for which LogitBoost obtains the best AUC among the benchmark learners (see Figure 2).

In addition to the AUC, Section D reports the accuracy obtained across all three methods. In binary classification problems, one metric is often insufficient to substantiate the classification capacity

Table 1: Maximum AUC achieved for ADNI cohorts using MLS, ACA-S, and ACA-L variants across all values of $M_{\text{res}}$, compared against the best benchmark (See Figure 2 for the best performing benchmark within each cohort). We also report the time needed to perform both the feature transformations and train the classifier for a single round of LPOCV. The AUCs reported across the different regimes within MLS and AUC demonstrate the sensitivity of different data sets to the type of SVM separating boundary. The difference in reported AUC for the ACA-S vs. ACA-L regimes also highlight the sensitivity of this dataset to the choice of residual subspace. Both the MLS and ACA methods are capable of elevating the AUC obtained by LogitBoost and SVM with linear separating boundary while also achieving a significant reduction in overall run time.

| Balanced | | | | | | | |
|---|---|---|---|---|---|---|---|
| Regime
SVM | MLS
Linear | MLS
RBF | ACA-S
Linear | ACA-S
RBF | ACA-L
Linear | ACA-L
RBF | Best
Benchmark |
| AD vs. CN | 0.838 | **0.894** | 0.821 | 0.883 | 0.782 | 0.864 | 0.789 |
| Time (ms) | 116.1 | 130.3 | **70.6** | 87.8 | 74.0 | 78.7 | 1173.42 |
| AD vs. LMCI | 0.750 | **0.886** | 0.863 | 0.863 | 0.647 | 0.860 | 0.790 |
| Time (ms) | 212.1 | 287.0 | **81.8** | 109.4 | 85.8 | 112.1 | 2302.33 |
| CN vs. LMCI | 0.923 | 0.937 | **0.970** | 0.968 | 0.830 | 0.909 | 0.910 |
| Time (ms) | 178.2 | 247.5 | 80.1 | 93.8 | **79.7** | 89.1 | 246.69 |
| Unbalanced | | | | | | | |
| Regime
SVM | MLS
Linear | MLS
RBF | ACA-S
Linear | ACA-S
RBF | ACA-L
Linear | ACA-L
RBF | Best
Benchmark |
| AD vs. CN | 0.865 | 0.910 | 0.875 | 0.912 | 0.864 | **0.913** | 0.789 |
| Time (ms) | 114.6 | 129.5 | **76.2** | 93.0 | 77.0 | 81.9 | 1173.42 |
| AD vs. LMCI | 0.743 | 0.883 | 0.860 | **0.889** | 0.743 | 0.883 | 0.790 |
| Time (ms) | 212.6 | 289.8 | **115.3** | 179.1 | 121.8 | 178.1 | 2302.33 |
| CN vs. LMCI | 0.927 | 0.938 | 0.955 | **0.959** | 0.928 | 0.938 | 0.910 |
| Time (ms) | 175.3 | 237.9 | **91.4** | 135.9 | 94.6 | 146.2 | 246.69 |

of a given method. Indeed, AUC can be sensitive to the interpolation method between thresholds Muschelli (2019). Furthermore, datasets with $N_{\mathbf{A}} \gg N_{\mathbf{B}}$ can yield a high AUC but a low accuracy.

This is exemplified in the CN vs. LMCI cohort (see Table 3). For this cohort, we have $N_{\mathbf{A}} = 346$ and $N_{\mathbf{B}} = 54$. An SVM with linear separating hypersurface obtains an AUC of 0.91 and an accuracy of only 0.71 on this dataset. With this in mind, we report that FINDER is also capable of elevating the accuracy of SVM classification compared to the benchmark learners on all three ADNI cohorts (see Table 4 and Figure 5).

Moreover, in Rehman et al. (2024) the authors propose several models that include a subset of the ADNI blood plasma proteins plus other features such as age, sex, education and the APOE4 gene. Although we used all of the proteomic blood plasma features, our results are significantly higher than those presented in Figure 2 in that paper.

## 3.2 DEFORESTATION DETECTION VIA RADAR AND OPTICAL REMOTE SENSING

For a second assessment of FINDER and to test its versatility, we applied the direct residual method to remote sensing/detection of deforestation using optical (Sentinel-2 Drusch et al. (2012)) and Synthetic Aperture Radar (SAR) (Sentinel-1 Torres et al. (2012)) data. We note that a similar approach was proposed in Lakhina et al. (2004) in the context of anomaly detection for network traffic.

These two datasets require us to consider two types of noise. SAR data contains noise due to the relatively weak sensors, but is unaffected by cloud cover. Optical data comes from higher quality and higher resolution sensors, but clouds can obstruct or completely block the ground, significantly reducing the amount of usable data. These datasets are critical in detecting deforestation, illegal logging, and quantifying the loss of carbon absorption in the atmosphere Initiative et al. (2016).

Our hybrid FINDER approach filters the SAR data, applies the direct residual method to the optical data, and combines the processed data in a complementary way that is very effective for highly cloudy regions such as the West African coast, Madagascar, and parts of the Amazon forest and

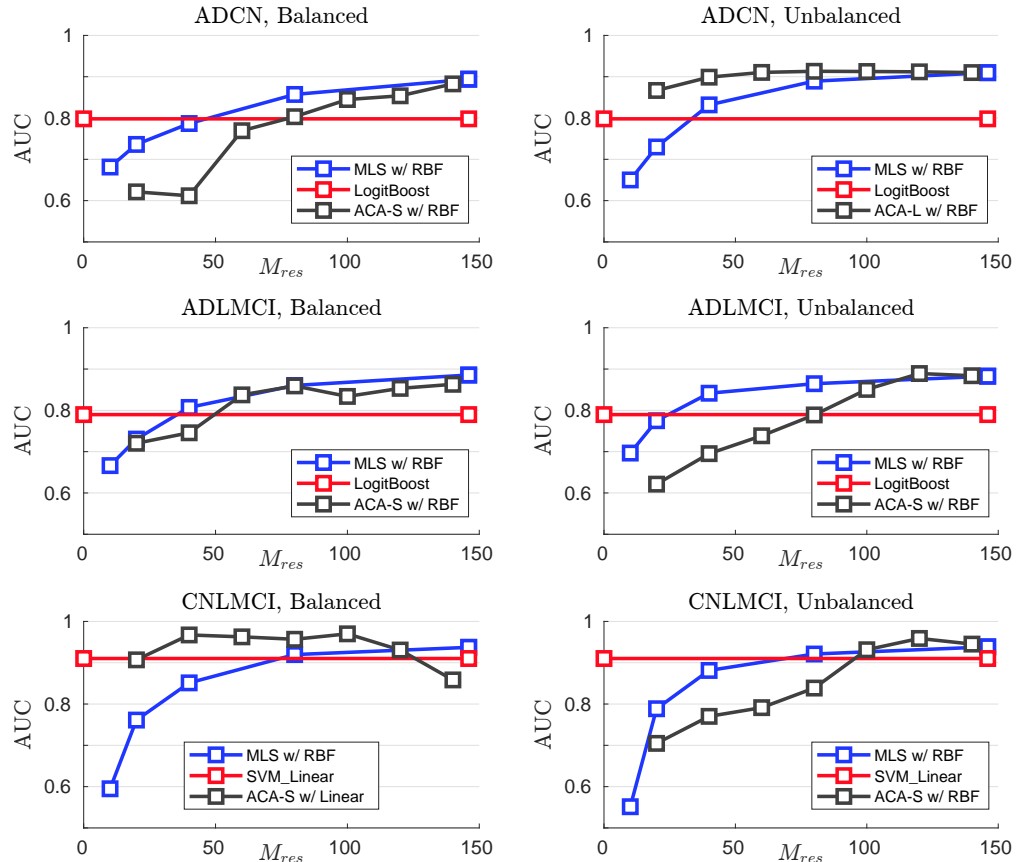

Figure 2: AUC obtained across all three methods for each of the three ADNI cohorts. Within each method (MLS, ACA, benchmark), the regime with the highest overall AUC obtained across all tested values of $M_{res}$ is reported. AUC can improve significantly from the benchmark level when both FINDER methods are employed with an RBF separating boundary. While the MLS method remains robust with respect to the choice to pre-balance or not pre-balanced the data, the ACA method is highly dependent on this choice. For the AD vs. CN cohort, the Unbalanced regime within ACA performs consistently better than benchmark and MLS. However, for the CN vs. LMCI cohort, the Balanced regime within ACA performs consistently better than benchmark and MLS. The data also demonstrate that the performance of FINDER is sensitive to the choice of $M_{res}$. The overall trend appears to be that larger $M_{res}$ achieve higher AUCs, though too large $M_{res}$ can diminish AUCs.

Southeast Asia Tang et al. (2023); Zhang et al. (2022). In particular, it significantly out-performs state-of-the-art methods such as Fusion Near Real Time (FNRT) Tang et al. (2023). This is made clear by the results in Table 2, where FINDER compares well against FNRT while using roughly 45 percent less data when FNRT can be used, and retains a fair proportion of its performance in the data scarce environments where FNRT is not applicable/usable.

We choose a test region of approximately 92 km × 92 km (corresponding to $9219 \times 9180$ pixels) in the Amazon forest. The Sentinel-2 optical bands are converted to a scalar valued Enhanced Vegetation Index (EVI) Huete et al. (2002) measurement on the terrain. This is a common measure in remote sensing to detect vegetation on land cover and can be used to detect loss of forest vegetation. Each optical pixel has a resolution of $10\,m \times 10\,m$. The optical Sentinel radar is resampled to same resolution as the optical EVI data and then filtered using a spatio-temporal Bayesian approach.

Using a simplified version of the residual eigenspace FINDER approach, an anomaly map is built from optical data that are fused with SAR with a Hidden Markov Model (HMM). The HMM then classifies the pixel as without change (forest), deforestation or cloud cover. For this experiment, the

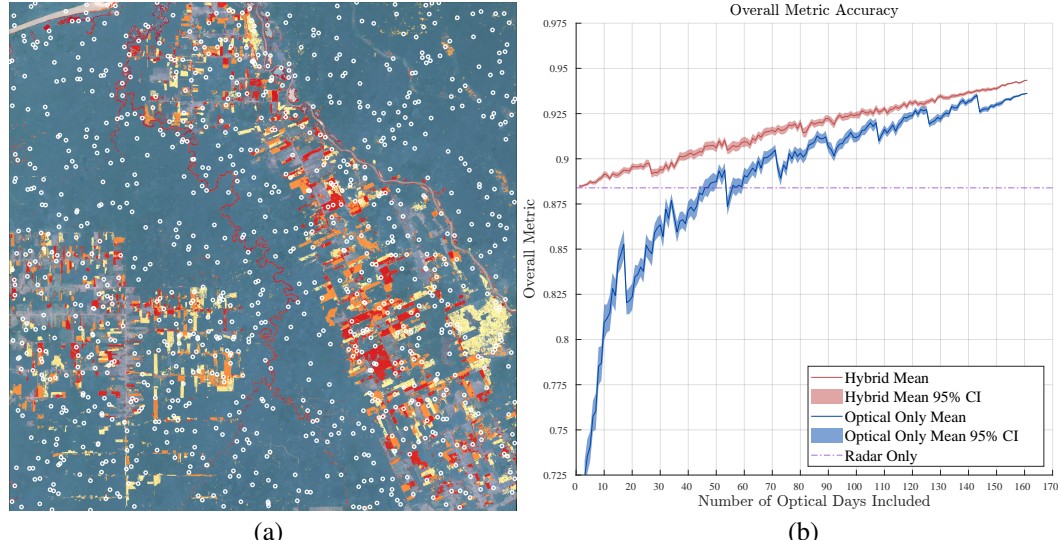

(a)                                        (b)

Figure 3: (a) Test region in the Amazon forest and validation samples. 1000 samples of the validation regions are selected. The region is formed by $9,219 \times 9,180$ pixels, each pixel a $10m \times 10m$ patch of land, representing the Enhanced Vegetation Index. The colored areas indicate the detection of deforestation with the Hybrid FINDER+HMM method by December 31 2022. (b) Overall metric accuracy with Hybrid and Optical only vs the number of available optical Sentinel-2 days.

FINDER spaces and parameters will be $\mathcal{H} = \mathcal{H}_M = \text{Span} \{\mathbf{e}_k\}_{k=1,\ldots,9219 \times 9180}$, where $e_k$ is a unit vector such that $\mathbf{e}_i \cdot \mathbf{e}_j = \delta[i - j]$, and $\mathcal{H}_{\text{res}} = \mathcal{H}_{\mathbf{A}}^{\perp}$ where $M_{\mathbf{A}} = 29$.

The performance of the algorithms is measured by selecting 1000 validation data points in the test region (See Figure 3(a)). Of these, 740 are stable forest with no change and in 260 the forest has been removed. The final state of the forest (stable or deforestation) at the end date is visually checked, and this is our ground truth.

The remote sensing community relies on three kinds of metrics to quantify detection algorithm performance: i) Overall accuracy refers to the percentage of correctly classified pixels (stable or deforestation) in the validation test area. ii) User's accuracy represents the probability that a pixel classified as a particular class (stable forest or deforestation) in a map is actually the correct validation state. iii) Producer's accuracy measures how well a map maker correctly identifies areas on the ground that belong to a specific class. Specific formulae from the remote sensing community can be found in Olofsson et al. (2014).

In Figure 3(b) the overall accuracy of the hybrid and optical-only methods is simulated for all possible numbers of available optical days. More specifically, for all $n \leq 161$ days we randomly choose 100 sets of $n$ Sentinel-2 EVI images and compute the hybrid and optical-only accuracies using the same sets, so as to get a direct comparison. The plot shows the mean and corresponding 95% confidence intervals from those 100 accuracies for each number of optical days. As this number decreases we clearly see that the optical-only accuracy reduces significantly. In contrast the hybrid method is robust to loss of optical data, limiting to the result from only using SAR data.

The remote sensing community has seen a recent surge in interest in combining optical and radar data to detect deforestation Chen et al. (2023). Table 2 summarizes the comparisons between the current state-of-the-art optical and radar hybrid approach and FINDER. The accuracy results presented are with common postprocessing remote sensing methods Chen et al. (2023) that give slightly higher accuracies for all the methods. We observe that FINDER achieves high accuracies for hybrid optical and radar data. For 71 days of optical training day an accuracy of 0.942 is achieved. If we reduce the training days to 35 the accuracy reduces only slightly to 0.933. In contrast, for the same training period with 71 days the performance of FNRT is poor. However, if we increase the training period for FNRT to 130 days (Jan 2018 to March 2020) the accuracy increases to 0.935.

Thus, we identify a strong use-case for FINDER here. Figure 3 and Table 2 show how well our method works under lack of optical data. Moreover, it can be used to track forest loss in almost real time. Many regions of the world at the risk of deforestation, such as large sections of the Amazons, coastal West Africa, Southeast Asia Tang et al. (2023); Zhang et al. (2022), etc, are too cloudy to supply FNRT methods with the data they need. In some cases, it can be years before sufficient data is assembled. Beyond the opportunity costs of not being able to act in time, this can also be a problem as deforestation during the data collection period can itself affect the performance of the algorithm.

**Remark 3.1.** *The necessity of applying FINDER to the optical data is discussed in Section D.2.*

Table 2: Accuracy results after postprocessing. Sentinel-2 optical data with HMM + FINDER has the same accuracy as joint optical and SAR data with the HMM + Finder. For the joint optical + SAR data the user accuracy is superior and the producer's accuracy is almost the same. Note that the state-of-the-art FNRT performs poorly when not enough optical training days exist, but improves significantly with more training data. Note that the timings for FNRT are 368 Google engine EECU-hours, which corresponds $\approx 3$ or $4$ wall hours. For the HMM + FINDER results the code was run using two 14-core 2.4 GHz Intel Xeon E5-2680v4 CPUs.

| Algorithm (Data) | Train Days | Overall Acc. | User | Producer | Comp Time (h) |
|---|---|---|---|---|---|
| FNRT (Hybrid) | 71 | 0.260 | 0.260 | 1.00[NA] | 368[#] |
| FNRT (Hybrid) | 130 | 0.935 | **0.892** | 0.707 | 368[#] |
| HMM + FINDER (Optical) | 71 | 0.936 | 0.801 | 0.748 | **13.95** |
| HMM + FINDER (Hybrid) | **35** | 0.933 | 0.839 | 0.718 | 49.34 |
| HMM + FINDER (Hybrid) | 71 | **0.942** | 0.865 | **0.752** | 49.47 |

## 4 LIMITATIONS AND CONCLUSION

FINDER is generic, versatile, robust in noisy regimes, and blends well with simple ML methods. However, its advantages diminish as noise decreases: stochastic features are computationally unnecessary in settings where data is "clean", "simple", or "ample" enough to easily learn the generalizing classes (see App. D.4 for an instructive examlpe).

Further, FINDER relies on judicious choices of the truncation parameters $M_{\mathbf{A}}$ and $M_{\text{res}}$. Empirically, the MLS method at least appears to demonstrate experimental predictability, with performance improving as $M_{\text{res}}$ is increased.

In contrast, the ACA methods demonstrate less obvious patterns in performance as $M_{\text{res}}$ is varied. Some heuristics for choosing $M_{\mathbf{A}}$ in particular are informed by Scree plots and the intuition of the user for their dataset. We are currently researching methods to make these choices, but it remains a stark weakness.

Another significant limitation lies in the Lemma 2.3 bounds being sub-optimal, as we eliminate the need to estimate $Y_r$ in our implementations to save computational costs. Furthermore, if the eigen-pairs $(\lambda_r^{\mathbf{A}}, \phi_r^{\mathbf{A}})$ and $(\lambda_r^{\mathbf{B}}, \phi_r^{\mathbf{B}})$ are too similar, then classification becomes nearly impossible.

Finally, FINDER is still only a binary classification regime: we offer no true breakthroughs on the question of multi-class problems, beyond decomposing them into a collection of costly binary problems.

Mitigating these weaknesses will remain the core focus of our future work. However, FINDER's initial results and applications are not just promising, but definitive evidence of its value.

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

## A  MATHEMATICAL DETAILS

We begin by assuming $\mathcal{H}$ is some arbitrary Hilbert space over the field $\mathbb{R}$ and $(\Omega, \mathcal{F}, \mathbb{P})$ is a complete probability space, substantially generalizing from the usual assumptions Schwab and Todor (2006). We will now derive Thm. 2.1 through a sequence of intermediate results. Let us first define:

## A.1 BOCHNER SPACES

**Definition A.1** (Simple functions). *We say that $v : \Omega \mapsto \mathcal{H}$ is simple if there exists a finite set of mutually disjoint measurable sets $\{E_n\}_{n=1}^N$ and vectors $\{e_n\}_{n=1}^N$ such that $v(\omega) = \sum_{n=1}^N \mathbb{I}_n(\omega)e_n$, where $\mathbb{I}_n(\omega) \equiv \mathbb{I}_{E_n}(\omega)$ are the indicator functions.*

**Definition A.2** (Bochner-measurable). *We say that $v : \Omega \mapsto \mathcal{H}$ is Bochner-measurable if there exists a sequence of simple functions $\{v_n\}_{n=1}^\infty$ such that for $\mathbb{P}$-a.e., we have $\lim_{n\to\infty} \|v_n(\omega) - v(\omega)\|_{\mathcal{H}} = 0$.*

Throughout this paper, we will assume the following conditions on $v$:

**Assumption A.1.** *$v(\omega)$ is a Bochner-measurable function from $\Omega$ to $\mathcal{H}$.*

**Assumption A.2.** *$\int_\Omega \|v(\omega)\|_{\mathcal{H}}^2 \, d\mathbb{P} < \infty$*

The set of all such $v : \Omega \to \mathcal{H}$ satisfying assumptions (A.1) - (A.2) is denoted $\mathcal{L}^2(\Omega, \mathcal{H})$, which constitutes a vector space under pointwise addition and scalar multiplication. The space $L^2(\Omega, \mathcal{H})$ comprises all the distinct equivalence classes in $\mathcal{L}^2(\Omega, \mathcal{H})$ where two functions are declared equivalent if they agree almost surely. We define an inner product on $L^2(\Omega, \mathcal{H})$ by $\langle u, v \rangle_{L^2(\Omega, \mathcal{H})} := \int_\Omega \langle u(\omega), v(\omega) \rangle_{\mathcal{H}} \, d\mathbb{P}$. With this inner product, $L^2(\Omega, \mathcal{H})$ gains the structure of a Hilbert space. The purpose of this section is to develop what is known as the *Karhunen-Loève (KL) expansion* of a random element $v$, given by

$$v(\omega) = \mathbb{E}(v) + \sum_{r=1}^R \lambda_r^{1/2} Y_r(\omega) \phi_r \tag{4}$$

where $R \in \mathbb{N} \cup \{\aleph_0\}$, $\mathbb{E}(v)$ is the expectation of $v$ (made precise later), $\{\lambda_r\}_{r=1}^R$ is a non-increasing sequence of positive real numbers with $\lambda_n \searrow 0$, $\{Y_r\}_{r=1}^R$ is an orthonormal set in $L^2(\Omega)$, and $\{\phi_r\}_{r=1}^R$ is an orthonormal set in $\mathcal{H}$. The proof of this expansion, which will be developed in this paper, relies on some elementary results about compact operators in Hilbert spaces.

By the Pettis theorem Diestel and Uhl (1977), $v : \Omega \to \mathcal{H}$ is Bochner-measurable if and only if $v$ is weakly measurable (i.e. the scalar-valued mapping $\omega \mapsto \langle v(\omega), e \rangle_{\mathcal{H}}$ is measurable for each $e \in \mathcal{H}$) and essentially separably valued (i.e. $v(\Omega_1)$ is separable for some $\Omega_1 \in \mathcal{F}$ with $\mathbb{P}(\Omega_1) = 1$). If $\mathcal{H}$ is separable (which we do not necessarily assume), then we need only verify that $v : \Omega \to \mathcal{H}$ is weakly measurable to assert its membership in $L^2(\Omega, \mathcal{H})$.

**Remark A.1.** *If $\mathcal{H} = L^2([a, b])$, then Assumption A.2 reduces to $\int_\Omega \int_a^b |v(x, \omega)|^2 dx \, d\mathbb{P} < \infty$ and the $L^2(\Omega, \mathcal{H})$ inner product is given as $\langle u, v \rangle_{L^2(\Omega, \mathcal{H})} = \int_\Omega \int_a^b v(x, \omega) u(x, \omega) dx \, d\mathbb{P}$.*

## A.2 HILBERT-SCHMIDT SPACES

Let $\mathcal{G}, \mathcal{H}$ be two Hilbert spaces over $\mathbb{R}$. Given $g \in \mathcal{G}$ and $h \in \mathcal{H}$, we may define a rank-one operator $\mathcal{H} \to \mathcal{G}$, symbolized by $g \otimes h$, given by $(g \otimes h)(x) = \langle h, x \rangle_{\mathcal{H}} g$ for all $x \in \mathcal{H}$.

We may define an inner product on two such operators as follows:

$$\langle g_1 \otimes h_1, g_2 \otimes h_2 \rangle := \langle g_1, g_2 \rangle_{\mathcal{G}} \langle h_1, h_2 \rangle_{\mathcal{H}} \tag{5}$$

The space $\mathrm{HS}(\mathcal{H}, \mathcal{G})$ is defined to be the Hilbert space formed by the closure of the linear span of the operators of the form $g \otimes h$ w.r.t. the inner product (5). If $\mathcal{G} = L^2(\Omega)$, we can embed $\mathrm{HS}(\mathcal{H}, L^2(\Omega))$ into $L^2(\Omega, \mathcal{H})$ via the correspondence $X \otimes e \mapsto Xe$. We prove that this inclusion is actually an isometric isomorphism by adapting (Hytönen et al., 2016, Prop. 1.4.4)

**Lemma A.1.** $L^2(\Omega, \mathcal{H})$ *is isometrically isomorphic to* $HS(\mathcal{H}, L^2(\Omega))$.

*Proof.* To first show that the inclusion $\iota : HS(\mathcal{H}, L^2(\Omega)) \to L^2(\Omega, \mathcal{H})$ is an isometry, let $X_1 \otimes e_1, X_2 \otimes e_2$ be two elements of $HS(\mathcal{H}, L^2(\Omega))$, whence we have:

$$
\begin{aligned}
\langle X_1 \otimes e_1, X_2 \otimes e_2 \rangle_{HS(\mathcal{H}, L^2(\Omega))} &= \langle X_1, X_2 \rangle_{L^2(\Omega)} \langle e_1, e_2 \rangle_{\mathcal{H}} \\
&= \int_\Omega \langle X_1(\omega) e_1, X_2(\omega) e_2 \rangle_{\mathcal{H}} \, d\mathbb{P} \\
&= \langle X_1 e_1, X_2 e_2 \rangle_{L^2(\Omega, \mathcal{H})}
\end{aligned}
$$

To show that $\iota$ is an isomorphism, it suffices to show that each $v \in L^2(\Omega, \mathcal{H})$ may be written as a potentially infinite sum $\sum_{j=1}^{N} X_j e_j$ for appropriate $X_j \in L^2(\Omega), e_j \in \mathcal{H}$. As $v$ is essentially separably valued, we may assume $v$ takes all of its values in a separable subset $Z \subseteq \mathcal{H}$. Letting $\{e_j\}_{j=1}^{N}$ ($N \in \mathbb{N} \cup \aleph_0$) be an orthonormal basis of $\overline{\text{Span}\{Z\}}$, we may expand $v(\omega)$ in its Fourier series with respect to the basis $\{e_j\}_{j=1}^{N}$ as follows

$$
v(\omega) := \sum_{j=1}^{N} X_j(\omega) e_j
$$

where $X_j(\omega) := \langle v(\omega), e_j \rangle_{\mathcal{H}}$. Each $X_j$ is measurable ($v$ is weakly measurable) and square summable (apply Cauchy-Schwarz inequality), and thus lies in $L^2(\Omega)$. If $N < \infty$, we are done. Otherwise, we must show that $\lim_{n \to \infty} \sum_{j=1}^{n} X_j e_j \to v$ in the $L^2(\Omega, \mathcal{H})$ norm. In this case, the remainder

$$
\left\| v(\omega) - \sum_{j=1}^{n} X_j(\omega) e_j \right\|_{\mathcal{H}}^2 \to 0 \text{ as } n \to \infty
$$

and is uniformly dominated by $\|v(\omega)\|_{\mathcal{H}}^2$ $\mathbb{P}$-a.e. The Dominated Convergence Theorem ensures that

$$
\left\| v - \sum_{j>n} X_j e_j \right\|_{L^2(\Omega, \mathcal{H})}^2 = \int_\Omega \left\| v(\omega) - \sum_{j>n} X_j(\omega) e_j \right\|_{\mathcal{H}}^2 \, d\mathbb{P} \to 0 \qquad \text{as} \qquad n \to \infty.
$$

$\square$

### A.3 BOCHNER INTEGRALS AND THEIR PROPERTIES

**Definition A.3** (Pettis integral). *Let $v \in L^2(\Omega, \mathcal{H})$. The Pettis integral is the unique $\mu \in \mathcal{H}$ for which* $\langle \mu, e \rangle_{\mathcal{H}} = \int_\Omega \langle v(\omega), e \rangle_{\mathcal{H}} \, d\mathbb{P}$ *for all $e \in \mathcal{H}$.*

**Definition A.4** (Bochner integral). *Let $v \in L^2(\Omega, \mathcal{H})$. We call $\mathbb{E}(v) := \int_\Omega v \, d\mathbb{P}$, the Bochner integral of $v$. Given an approximating sequence of simple functions $\{v_n\}_{n \in \mathbb{N}}$, the Bochner integral of $v$ is the limiting value of $\int_\Omega v_n(\omega) \, d\mathbb{P}$ as $n \to \infty$ (provided it exists).*

With these preliminaries in mind, we can now prove Theorem 2.1

*Proof.* Since all Hilbert spaces satisfy the Radon-Nikodým Property (RNP) and $v$ is Bochner integrable, it is also Pettis integrable and the two integrals coincide (Diestel and Uhl, 1977, Ch. 2). Thus

we may instead consider the element $v_0 := v - \mathbb{E}(v)$, which also lies in $L^2(\Omega, \mathcal{H})$. By our identification of $L^2(\Omega, \mathcal{H})$ with $\mathrm{HS}(\mathcal{H}, L^2(\Omega))$, there exists a unique Hilbert-Schmidt corresponding to $v_0$, which we will denote $H_{v_0}$. As all Hilbert-Schmidt operators are compact, we may invoke the spectral theorem to write the SVD of $H_{v_0}$ as $\sum_{r=1}^{R} \lambda_r^{1/2} Y_r \otimes \phi_r$ where $R \in \mathbb{N} \cup \{\aleph_0\}$. $\left\{\lambda_r^{1/2}\right\}_{r=1}^{R}$ comprises the non-increasing real-valued sequence of singular values of $H_{v_0}$ with $\lambda_r \searrow 0$. $\{Y_r\}_{r=1}^{R}$ comprises the left singular vectors of $H_{v_0}$ and forms an orthonormal set in $L^2(\Omega)$. $\{\phi_r\}_{r=1}^{R}$ comprises the right singular vectors of $H_{v_0}$ and forms an orthonormal set in $\mathcal{H}$. Using the correspondence once more, we see

$$v_0 = v - \mathbb{E}(v) = \sum_{r=1}^{R} \lambda_r^{1/2} Y_r \phi_r \qquad v_0 \in L^2(\Omega, \mathcal{H})$$

$\square$

## A.4 THE COVARIANCE OPERATOR

For each $v \in L^2(\Omega, \mathcal{H})$ the operator $H_{v_0}$ may be constructed as the mapping $e \mapsto \langle v - \mathbb{E}(v), e \rangle_{\mathcal{H}}$. We define the *covariance operator* of $v$ by $H_{v_0}^* H_{v_0}$ and denote it $\mathcal{C}_v$. We immediately obtain

$$\mathcal{C}_v(e) = \mathbb{E}(\langle v - \mathbb{E}(v), e \rangle_{\mathcal{H}} (v - \mathbb{E}(v)))$$

for each $e \in \mathcal{H}$. We obtain the sequences $\{\lambda_r\}_{r=1}^{R}$ and $\{\phi_r\}_{r=1}^{R}$ in the KL expansion of $v$ as the descending sequence of eigenvalues and eigenvectors of $\mathcal{C}_v$. The sequence $\{Y_r\}_{r=1}^{R}$ comprises the left singular vectors of $H_{v_0}$, obtained as

$$Y_r := \lambda_r^{-1/2} H_{v_0} \phi_r$$

Furthermore, each $Y_r$ has expectation zero, which results from the fact that the Bochner (and thus Pettis) integral of $v - \mathbb{E}(v)$ is zero. Indeed, for any $s \leq R$, we have

$$0 = \left\langle \lambda_s^{-1/2} \phi_s, \int_\Omega v(\omega) - \mathbb{E}(v) \, d\mathbb{P} \right\rangle_{\mathcal{H}}$$
$$= \int_\Omega \left\langle \lambda_s^{-1/2} \phi_s, v(\omega) - \mathbb{E}(v) \right\rangle_{\mathcal{H}} d\mathbb{P}$$
$$= \int_\Omega Y_s(\omega) \, d\mathbb{P}$$

**Remark A.2.** *If $\mathcal{H} = L^2([a, b])$, then $\mathbb{E}(v) = \int_\Omega v(-, \omega) \, d\mathbb{P}$. Furthermore, the covariance operator associated to $v$ is a kernel operator whose kernel is*

$$\mathcal{K}(x, y) = \int_\Omega [v(x, \omega) - \mathbb{E}((v(x, \omega))][v(y, \omega) - \mathbb{E}((v(y, \omega))] \, d\mathbb{P},$$

*that is, for any $f \in L^2([a, b])$, $C_v(f)(x) = \int_a^b \mathcal{K}(x, y) f(y) dy$.*

## A.5 COMPUTABLE SUBSPACES

For the purposes of computation, only finite dimensional Hilbert spaces are admissible. Suppose $\mathcal{H}_M \leq \mathcal{H}$ is such a finite dimensional subspace and $P_{\mathcal{H}_M}$ is the projection onto $\mathcal{H}_M$ operator. A desirable quality of FINDER, or any machine learning approach, would be that this finite dimensional digitization that occurs when replacing $\mathcal{H}$ with $\mathcal{H}_M$ also translates to replacing $L^2(\Omega, \mathcal{H})$ with $L^2(\Omega, \mathcal{H}_M)$ and $\mathrm{HS}(\mathcal{H}, L^2(\Omega))$ with $\mathrm{HS}(\mathcal{H}_M, L^2(\Omega))$. Mathematically, we would like to verify that $P_{\mathcal{H}_M}$ induces maps $P' : L^2(\Omega, \mathcal{H}) \to L^2(\Omega, \mathcal{H}_M)$ and $P'' : \mathrm{HS}(\mathcal{H}, L^2(\Omega)) \to \mathrm{HS}(\mathcal{H}_M, L^2(\Omega))$ such that the following diagram commutes:

$$L^2(\Omega, \mathcal{H}) \xrightarrow{\quad P' \quad} L^2(\Omega, \mathcal{H}_M)$$
$$\downarrow \eta \qquad\qquad\qquad \downarrow \eta$$
$$\mathrm{HS}(\mathcal{H}, L^2(\Omega)) \xrightarrow{\quad P'' \quad} \mathrm{HS}(\mathcal{H}_M, L^2(\Omega))$$

where $\eta$ is the correspondence $Xe \leftrightarrow X \otimes e$. In the language of category theory, if $L^2(\Omega, -)$ and $\mathrm{HS}(-, L^2(\Omega))$ are functors from the category of Hilbert spaces over $\mathbb{R}$ to itself, then $\eta$ should act as a natural isomorphism between these two functors.

*Proof.* In fact, we put $P'$ as the map $v \mapsto P_{\mathcal{H}_M} v$ and $P''$ as the map $H \mapsto H P_{\mathcal{H}_M}$ and compute for each element of the form $Xe \in L^2(\Omega, \mathcal{H})$

$$\eta(P'(Xe)) = \eta(X(P_{\mathcal{H}_M} e)) = X \otimes (P_{\mathcal{H}_M} e) = P''(X \otimes e) = P'' \eta(Xe)$$

$\square$

## B APPLICATIONS TO BINARY CLASSIFICATION

Consider two random elements of $L^2(\Omega, \mathcal{H})$, $v^{\mathbf{A}}$ and $v^{\mathbf{B}}$, whose instantiations $v^{\mathbf{A}}(\omega)$ and $v^{\mathbf{B}}(\omega)$ are thought of as belonging to Class $\mathbf{A}$ or Class $\mathbf{B}$. This section presents a method for determining whether or not an observed random element $u \in \mathcal{H}$ belongs to Class $\mathbf{A}$ or Class $\mathbf{B}$.

**Definition B.1** (Equality (in distribution)). *$v, \tilde{v} \in L^2(\Omega, \mathcal{H})$ are said to be equal in distribution, denoted $v \stackrel{d}{=} \tilde{v}$, if $\Pr(v \in B) = \Pr(\tilde{v} \in B)$ for all Borel-measurable subsets $B$ of $\mathcal{H}$.*

**Remark B.1.** *We can restrict to the cases where $B$ is an open set in $\mathcal{H}$.*

**Corollary B.1.** *Let $v \stackrel{d}{=} \tilde{v}$. Then $\mathcal{C}_v$ and $\mathcal{C}_{\tilde{v}}$ share the same spectrum, i.e., $\{\lambda_r\}_{r=1}^R = \left\{\tilde{\lambda}_r\right\}_{r=1}^R$. Further, $\ker(\mathcal{C}_v - \lambda_r I) = \ker(\mathcal{C}_{\tilde{v}} - \tilde{\lambda}_r I))$ for each $r = 1, 2, \ldots R$.*

Note the converse need not be true; if $\mathcal{C}_v = \mathcal{C}_{\tilde{v}}$, the it need not be the case that $v \stackrel{d}{=} \tilde{v}$. In fact, if $\mathcal{C}_v$ possesses eigen-pairs $\{\lambda_r, \phi_r\}_{r=1}^R$, then for any zero mean, orthonormal sequence $\left\{\tilde{Y}_r(\omega)\right\}_{r=1}^R \subseteq L^2(\Omega)$, the random element $\tilde{v} := \sum_{r=1}^R \lambda_r^{1/2} \tilde{Y}_r(\omega) \phi_r$ admits covariance operator $\mathcal{C}_{\tilde{v}}$ equal to $\mathcal{C}_v$.

Although knowledge of the sequence of singular values $\left\{\lambda_r^{1/2}\right\}_{r=1}^R$, right singular vectors $\{\phi_r\}_{r=1}^R$, and left singular vectors (random variables) $\{Y_r\}_{r=1}^R$ are all needed to fully describe the distribution of the random element $v$, we describe a classification method which does not require knowledge of the sequence of random variables $\{Y_r\}_{r=1}^R$ present in the KL expansion of $v$. This is particularly advantageous, since the $\{Y_r\}_{r=1}^R$ are not necessarily independent, just uncorrelated, thus estimation of the joint distribution of $\{Y_r\}_{r=1}^R$ requires a high dimensional estimation problem which is quite difficult to do from even a moderately sized empirical dataset $\{v_i\}_{i=1}^N$ without additional assumptions on $v$.

In contrast, the estimation of the eigen-pairs $\left\{\widehat{\lambda}_r^{1/2}, \widehat{\phi}_r\right\}_{r=1}^R$ from the empirical covariance operator $\widehat{\mathcal{C}_v} := \frac{1}{N-1} \sum_{i=1}^N (v_i - \overline{v}) \otimes (v_i - \overline{v})$ (with $\overline{v} := \frac{1}{N} \sum_{i=1}^N v_i$) is relatively easy. In particular, if $\mathcal{H} = \mathbb{R}^P$, then $\widehat{\mathcal{C}_v}$ is just the empirical covariance matrix, and the eigen-pairs $\left\{\widehat{\lambda}_r^{1/2}, \widehat{\phi}_r\right\}_{r=1}^R$ are just the square roots of the eigenvalues and the eigenvectors of this matrix.

## B.1 CLASS CONCENTRATION BOUNDS

Throughout the remainder of this paper, assume that $\mathbb{E}(v^{\mathbf{A}}) = 0$, since we can impose this by setting $v^{\mathbf{A}} \to v^{\mathbf{A}} - \mathbb{E}(v^{\mathbf{A}})$ and $v^{\mathbf{B}} \to v^{\mathbf{B}} - \mathbb{E}(v^{\mathbf{A}})$. Let $v^{\mathbf{A}}$ and $v^{\mathbf{B}}$ have KL expansions:

$$v^{\mathbf{A}} = \sum_{r=1}^{R_{\mathbf{A}}} \lambda_r^{\mathbf{A}\,1/2} Y_r^{\mathbf{A}} \phi_r^{\mathbf{A}}$$

$$v^{\mathbf{B}} = \mathbb{E}(v^{\mathbf{B}}) + \sum_{r=1}^{R_{\mathbf{B}}} \lambda_r^{\mathbf{B}\,1/2} Y_r^{\mathbf{B}} \phi_r^{\mathbf{B}}$$

We will also make use of following corollaries of Hille's theorem (Sullivan, 2024, Thm. 1) throughout:

**Lemma B.1.** *Let $K : \mathcal{H} \to \mathcal{H}$ be a bounded linear operator. For any $v \in L^2(\Omega, \mathcal{H})$, we have that:*

$$\mathbb{E}(Kv) = K\mathbb{E}(v), \qquad \mathcal{C}_{Kv} = K\mathcal{C}_v K^*$$

The method of classification consists of computing an orthonormal basis $\{s_m\}_{m=1}^{M_{\mathrm{res}}}$ of a finite dimensional subspace $\mathcal{H}_{\mathrm{res}} \leq \mathcal{H}_M$ for which the projections $P_{\mathcal{H}_{\mathrm{res}}} v^{\mathbf{A}}$ and $P_{\mathcal{H}_{\mathrm{res}}}(v^{\mathbf{B}} - \mathbb{E}(v^{\mathbf{B}}))$ (which equals by $P_{\mathcal{H}_{\mathrm{res}}} v^{\mathbf{B}} - \mathbb{E}(P_{\mathcal{H}_{\mathrm{res}}} v^{\mathbf{B}})$ Lemma B.1) concentrate in distinct regions of $\mathcal{H}$ with high probability (recall that we've assumed that $\mathbb{E}(v^{\mathbf{A}})$ has been pre-subtracted from both $v^{\mathbf{A}}$ and $v^{\mathbf{B}}$). To this end, we establish the following bounds for the quantities $\int_\Omega \|P_{\mathcal{H}_{\mathrm{res}}} v^{\mathbf{A}}\|_{\mathcal{H}}^2 \, d\mathbb{P}$ and $\int_\Omega \|P_{\mathcal{H}_{\mathrm{res}}}(v^{\mathbf{B}} - \mathbb{E}(v^{\mathbf{B}}))\|_{\mathcal{H}}^2 \, d\mathbb{P}$.

$$
\begin{aligned}
\int_\Omega \|P_{\mathcal{H}_{\mathrm{res}}} v^{\mathbf{A}}\|_{\mathcal{H}}^2 \, d\mathbb{P} &= \int_\Omega \sum_{m=1}^{M_{\mathrm{res}}} \left\langle s_m, v^{\mathbf{A}} \right\rangle_{\mathcal{H}}^2 \, d\mathbb{P} \\
&= \sum_{m=1}^{M_{\mathrm{res}}} \int_\Omega \left\langle \left\langle s_m, v^{\mathbf{A}} \right\rangle_{\mathcal{H}} v^{\mathbf{A}}, s_m \right\rangle_{\mathcal{H}} \, d\mathbb{P} \\
&= \sum_{m=1}^{M_{\mathrm{res}}} \left\langle \mathbb{E}(\left\langle s_m, v^{\mathbf{A}} \right\rangle_{\mathcal{H}} v^{\mathbf{A}}), s_m \right\rangle_{\mathcal{H}} \\
&= \sum_{m=1}^{M_{\mathrm{res}}} \left( \left\langle \mathcal{C}_{v^{\mathbf{A}}} s_m, s_m \right\rangle_{\mathcal{H}} \right) \\
&= \sum_{m=1}^{M_{\mathrm{res}}} \left( \left\langle \sum_{r=1}^{R_{\mathbf{A}}} \lambda_r^{\mathbf{A}} \left\langle s_m, \phi_r^{\mathbf{A}} \right\rangle_{\mathcal{H}} \phi_r^{\mathbf{A}}, s_m \right\rangle_{\mathcal{H}} \right) \\
&= \sum_{m=1}^{M_{\mathrm{res}}} \sum_{r=1}^{R_{\mathbf{A}}} \lambda_r^{\mathbf{A}} \left\langle \phi_r^{\mathbf{A}}, s_m \right\rangle_{\mathcal{H}}^2
\end{aligned}
$$

A similar calculation proves that $\int_\Omega \|P_{\mathcal{H}_{\mathrm{res}}}(v^{\mathbf{B}} - \mathbb{E}(v^{\mathbf{B}}))\|_{\mathcal{H}}^2 \, d\mathbb{P} = \sum_{m=1}^{M_{\mathrm{res}}} \sum_{r=1}^{R_{\mathbf{B}}} \lambda_r^{\mathbf{B}} \left\langle \phi_r^{\mathbf{B}}, s_m \right\rangle_{\mathcal{H}}^2$. Hence using Markov's (or Chebyshev's first) inequality Stein and Shakarchi (2005), we have for any $\varepsilon > 0$

$$\Pr(\|P_{\mathcal{H}_{\mathrm{res}}} v^{\mathbf{A}}\|_{\mathcal{H}}^2 \geq \varepsilon^2) \leq \varepsilon^{-2} \sum_{m=1}^{M_{\mathrm{res}}} \sum_{r=1}^{R_{\mathbf{A}}} \lambda_r^{\mathbf{A}} \left\langle \phi_r^{\mathbf{A}}, s_m \right\rangle_{\mathcal{H}}^2 \tag{6}$$

and for Class $\mathbf{B}$

$$\Pr(\|P_{\mathcal{H}_{\text{res}}}(v^{\mathbf{B}} - \mathbb{E}(v^{\mathbf{B}}))\|_{\mathcal{H}}^2 \geq \varepsilon^2) \leq \varepsilon^{-2} \sum_{m=1}^{M_{\text{res}}} \sum_{r=1}^{R_{\mathbf{B}}} \lambda_r^{\mathbf{B}} \left\langle \phi_r^{\mathbf{B}}, s_m \right\rangle_{\mathcal{H}}^2 \qquad (7)$$

The two inequalities (6) and (7) imply that an upper bound for the probability that a random element $v$ equal in distribution to $v^{\mathbf{A}}$ lies inside a ball centered at the origin with probability proportional to $\int_{\Omega} \|P_{\mathcal{H}_{\text{res}}} v^{\mathbf{A}}\|_{\mathcal{H}}^2 \, d\mathbb{P}$ and that $v$ equal in distribution to $v^{\mathbf{B}}$ lies inside a ball centered at $P_{\mathcal{H}_{\text{res}}} \mathbb{E}(v^{\mathbf{B}})$ with probability proportional to $\int_{\Omega} \|P_{\mathcal{H}_{\text{res}}}(v^{\mathbf{B}} - \mathbb{E}(v^{\mathbf{B}}))\|_{\mathcal{H}}^2 \, d\mathbb{P} = \int_{\Omega} \|P_{\mathcal{H}_{\text{res}}} v^{\mathbf{B}} - \mathbb{E}(P_{\mathcal{H}_{\text{res}}} v^{\mathbf{B}})\|_{\mathcal{H}}^2 \, d\mathbb{P}$

## B.2 Multilevel Subspaces (MLS)

The approach used in the MLS regime within FINDER endeavors to produce a subspace $\mathcal{H}_{\text{res}}$ for which the quantity $\sum_{m=1}^{M_{\text{res}}} \sum_{r=1}^{R_{\mathbf{A}}} \lambda_r^{\mathbf{A}} \left\langle \phi_r^{\mathbf{A}}, s_m \right\rangle_{\mathcal{H}}^2$ is small. For then, when the sampled features $v_i^{\mathbf{A}}$ are updated as $v_i^{\mathbf{A}} \to P_{\mathcal{H}_{\text{res}}} v_i^{\mathbf{A}}$, they will concentrate about the origin with high probability. An SVM (with an RBF kernel, if desired) is suitable to binary classification problems where the data is assumed to possess some sort of geometric separation; this method can reliably classify $\mathbf{A}$ from $\mathbf{B}$.

In fact, a sequence of nested subspaces $\{\mathbb{V}_\ell\}_{\ell=0}^L$ ($L \in \mathbb{N} \cup \{\aleph_0\}$) is created to this end. To say that the sequence of subspaces is nested means that $\mathbb{V}_\ell \subseteq \mathbb{V}_{\ell+1}$ ($\ell \in \mathbb{N}$), and $\overline{\bigcup_{\ell=0}^L \mathbb{V}_\ell} = \mathcal{H}$. Of course, in computation $L$ will be finite. Define $\mathbb{W}_\ell$ to be the orthogonal complement of $\mathbb{V}_{\ell-1}$ in $\mathbb{V}_\ell$, which makes it so that $\mathcal{H} = \overline{\bigcup_{\ell=0}^L \mathbb{V}_\ell} = \mathbb{V}_0 \oplus \bigoplus_{\ell=1}^L \mathbb{W}_\ell$.

We introduce an integer $M_{\mathbf{A}}$ between 1 and $R_{\mathbf{A}}$, thinking of it as a "truncation parameter" for $v^{\mathbf{A}}$. Let $\mathcal{H}_{\mathbf{A}} := \text{Span} \left\{ \phi_r^{\mathbf{A}} \right\}_{r=1}^{M_{\mathbf{A}}}$. The MLS approach within FINDER sets $\mathbb{V}_0 = \mathcal{H}_{\mathbf{A}}$. The subspaces $\mathbb{V}_\ell$ are chosen such that each $\mathbb{W}_\ell$ are finite dimensional ($\mathbb{W}_\ell = \text{Span} \left\{ \phi_k^\ell \right\}_{k=1}^{M_\ell}, \phi_k^\ell \in \mathcal{H}$). The $\phi_k^\ell$ are constructed to satisfy the orthonormality condition: $\left\langle \phi_{k_1}^{\ell_1}, \phi_{k_2}^{\ell_2} \right\rangle_{\mathcal{H}} = \delta[k_1 - k_2]\delta[\ell_1 - \ell_2]$. For any choice of $\ell$, $\mathbb{V}_\ell$ satisfies the inequality

$$\Pr(\|P_{\mathbb{V}_\ell} v^{\mathbf{A}}\|_{\mathcal{H}}^2 \geq \varepsilon^2) \leq \varepsilon^{-2} \sum_{r > M_{\mathbf{A}}} \lambda_r^{\mathbf{A}} \qquad (8)$$

The MLS approach is tractable when $\mathcal{H} = L^2(U)$ for some $U \subseteq \mathbb{R}^D$. In this context, the $\phi_k^\ell$ are chosen to be characteristic functions whose support is constructed by a kd-tree division of $U$ (where k = D). In computation, the $\phi_k^\ell$ are chosen to be vectors in $\mathbb{R}^F$ with relatively few nonzero entries, which lends well to efficient memory usage. Explicit construction of the $\phi_k^\ell$, the choice of $M_\ell$ and $L$ are an adaption of a construction detailed in Tausch and White (2003).

## B.3 Anomalous Class Adapted Spaces

The Anomalous Class Adapted (ACA) approach differs from the MLS approach in that the latter does not consider the specific eigen-structure of $\mathcal{C}_{v^{\mathbf{B}}}$, only that it is assumed to be distinct from the eigen-structure of $\mathcal{C}_{v^{\mathbf{A}}}$. Our choice of constructing such a finite-dimensional subspace $\mathcal{H}_{\text{res}}$ endeavors to make both the RHS of (6) and (7) "small" such that $P_{\mathcal{H}_{\text{res}}} v^{\mathbf{A}}$ and $P_{\mathcal{H}_{\text{res}}} v^{\mathbf{B}}$ lie in separate parts of $\mathcal{H}$ with high probability.

### B.3.1 ACA -S

Like the MLS approach, we consider $\mathcal{H}_{\text{res}}$ to be a subspace of $\mathcal{H}_{\mathbf{A}}^{\perp}$. Then the first $M_{\mathbf{A}}$ terms in (6) vanish such that

$$\Pr(\|P_{\mathcal{H}_{\text{res}}} v^{\mathbf{A}}\|_{\mathcal{H}}^2 \geq \varepsilon^2) \leq \varepsilon^{-2} \sum_{m=1}^{M_{\text{res}}} \sum_{r=M_{\mathbf{A}}+1}^{R_{\mathbf{A}}} \lambda_r^{\mathbf{A}} \left\langle \phi_r^{\mathbf{A}}, s_m \right\rangle_{\mathcal{H}}^2 \leq \varepsilon^{-2} \sum_{r=M_{\mathbf{A}}+1}^{R_{\mathbf{A}}} \lambda_r^{\mathbf{A}} \qquad (9)$$

To minimize the RHS of (7), subject to $\mathcal{H}_{\text{res}} \leq \mathcal{H}_{\mathbf{A}}^{\perp}$, let $\{\psi_i\}_{i \in I}$ be an orthonormal basis for $\mathcal{H}_{\mathbf{A}}^{\perp}$, let $V_{M_{\mathbf{A}}} : \ell^2(I) \to \mathcal{H}_{\mathbf{A}}^{\perp}$ be the isometry $V_{M_{\mathbf{A}}} f = \sum_{i \in I} f(i)\psi_i$, whence

$$\min_{s_m} \sum_{m=1}^{M_{\text{res}}} \sum_{r=1}^{R_{\mathbf{B}}} \lambda_r^{\mathbf{B}} \left\langle \phi_r^{\mathbf{B}}, s_m \right\rangle_{\mathcal{H}}^2 = \min_{s_m} \sum_{m=1}^{M_{\text{res}}} \left\langle s_m, \mathcal{C}_{v^{\mathbf{B}}} s_m \right\rangle_{\mathcal{H}} = \min_{t_m} \sum_{m=1}^{M_{\text{res}}} \left\langle t_m, V_{M_{\mathbf{A}}}^* \mathcal{C}_{v^{\mathbf{B}}} V_{M_{\mathbf{A}}} t_m \right\rangle_{\mathcal{H}}.$$
$$(10)$$

where $t_m = V_{M_{\mathbf{A}}}^* s_m$. It follows that the minimum occurs when $t_m$ are the eigenvectors corresponding to the smallest $M_{\text{res}}$ eigenvalues of $V_{M_{\mathbf{A}}}^* \mathcal{C}_{v^{\mathbf{B}}} V_{M_{\mathbf{A}}}$. Then we set $s_m = V_{M_{\mathbf{A}}} t_m$.

*Remark*: If $\mathcal{H} = \mathbb{R}^F$, then let $\{\phi_r^{\mathbf{A}}\}_{r=1}^{F}$ denote orthonormalized eigenvectors of the matrix $\mathcal{C}_{v^{\mathbf{A}}}$. Then $V_{M_{\mathbf{A}}} = \begin{bmatrix} \phi_{M_{\mathbf{A}}+1}^{\mathbf{A}} & \cdots & \phi_F^{\mathbf{A}} \end{bmatrix}$, and we set the vectors $\{t_m\}_{m=1}^{M_{\text{res}}}$ as the unit eigenvectors corresponding to the $M_{\text{res}}$ smallest eigenvalues of the matrix $V_{M_{\mathbf{A}}}^{\top} \mathcal{C}_{v^{\mathbf{B}}} V_{M_{\mathbf{A}}}$. Note that $t_m \in \mathbb{R}^{F-M_{\mathbf{A}}}$. Then we set $s_m = V_{M_{\mathbf{A}}} t_m$ and $\mathcal{H}_{\text{res}} = \text{Span}\{s_m\}_{m=1}^{M_{\text{res}}}$. This choice of $\mathcal{H}_{\text{res}}$ constitutes the ACA-S method of FINDER.

Lemma B.1 offers a simple generalization of a property of random vectors: namely if $v \in L^2(\Omega, \mathbb{R}^F)$ is a random vector and $K \in \mathbb{R}^{q \times F}$ is an arbitrary matrix then the covariance matrix of $Kv$ is simply $K\mathcal{C}_v K^{\top}$. From Lemma (B.1), it follows that upon projection onto $\mathcal{H}_{\text{res}}$, the covariance operators update as $\mathcal{C}_{v^{\mathbf{C}}} \to P_{\mathcal{H}_{\text{res}}} \mathcal{C}_{v^{\mathbf{C}}} P_{\mathcal{H}_{\text{res}}}^*$. In computation, the operator $Q_{\text{res}} := \sum_{m=1}^{M_{\text{res}}} \mathbf{e}_m \otimes s_m$ is used in place of $P_{\mathcal{H}_{\text{res}}}$, where $\mathbf{e}_m$ is the $m$th standard basis vector of $\mathbb{R}^{M_{\text{res}}}$

### B.3.2   ACA-L

The tendency of the projected $v_i$ to concentrate in different regions of $\mathcal{H}$ makes an SVM with RBF kernel a particularly attractive choice of machine, though we test a linear kernel as well. However, if the within-class spread of the two classes is large, relative to the distance between the class means, then the above projection $Q_{\text{res}}$ may fail to reliably separate the two classes. If it is the case that the two class means are particularly close, it may be more desirable to choose a subspace $\mathcal{H}_{\text{res}}$ with the property that $Q_{\text{res}} v^{\mathbf{A}}$ concentrates about the origin with low spread and $Q_{\text{res}} v^{\mathbf{B}}$ also concentrates about the origin but with large spread, as in Figure 1. Since concentration bound (9) holds as long as $\mathcal{H}_{\text{res}} \leq \mathcal{H}_{\mathbf{A}}^{\perp}$, one can instead set the $t_m$ to be the eigenvectors of $V_{M_{\mathbf{A}}}^* \mathcal{C}_{v^{\mathbf{B}}} V_{M_{\mathbf{A}}}$ corresponding to the *largest* eigenvalues of this operator, thereby *maximizing* $\|P_{\mathcal{S}}(v^{\mathbf{B}} - \mathbb{E}(v^{\mathbf{B}}))\|_{L^2(\Omega, \mathcal{H})}^2$ over all possible subspaces of dimension $M_{\text{res}}$. This choice of $\mathcal{H}_{\text{res}}$ constitutes the ACA-L method of FINDER. Furthermore, we employ several, equally spaced values of $M_{\text{res}}$ in experimentation.

### B.4   COMPLEXITY ANALYSIS FOR FEATURE CONSTRUCTION

Suppose that class $\mathbf{A} \in \mathbb{R}^{F \times N_{\mathbf{A}}}$ consists of $F$ features and $N_{\mathbf{A}}$ samples. Similarly $\mathbf{B} \in \mathbb{R}^{F \times N_{\mathbf{B}}}$ with $F$ features and $N_{\mathbf{A}}$ samples. Suppose we have $M_{\mathbf{A}}$ principal components truncation parameters of Class $\mathbf{A}$. There are three basic modules for computing the training data: (i) KL module that computes the covariance matrix. (ii) Construction module that constructs the basis for the residual subspace. This module varies according to the type of residual subspace of the basis that is constructed. (iii) Feature module that computes the projection on the residual subspace and thus the novel features.

Direct calculations estimate the nominal computational complexity of each FINDER variant as:

**Direct residual subspace:** (i) KL and Construction modules: If $N_{\mathbf{A}} < F$ then $\mathcal{O}(N_{\mathbf{A}}^2 M_{\mathbf{A}}) + \mathcal{O}(F N_{\mathbf{A}}^2 M_{\mathbf{A}})$ else $\mathcal{O}(F^2 M_{\mathbf{A}})$. (ii) Feature module: $\mathcal{O}(F N_{\mathbf{B}} M_{\mathbf{A}})$

**MLS:** (i) KL module: If $N_{\mathbf{A}} < F$ then $\mathcal{O}(N_{\mathbf{A}}^2 M_{\mathbf{A}}) + \mathcal{O}(F N_{\mathbf{A}}^2 M_{\mathbf{A}})$ else $\mathcal{O}(F^2 M_{\mathbf{A}})$.
(ii) Construction module: $\approx \mathcal{O}(F \log F)$ (iii) Feature module: $\approx \mathcal{O}((N_{\mathbf{A}} + N_{\mathbf{B}}) F \log F))$

**ACA:** (i) KL & Construction modules: $\mathcal{O}(F \log F)$. (ii) Feature module: $\mathcal{O}((F - M_{\mathbf{A}}) F (N_{\mathbf{A}} + N_{\mathbf{B}}))$.

## C TESTING

### C.1 ESTIMATES

For LPOCV, the datasets $\left\{v_i^{\mathbf{A}}\right\}_{i=1}^{N_{\mathbf{A}}}$, $\left\{v_i^{\mathbf{B}}\right\}_{i=1}^{N_{\mathbf{B}}}$ are both divided into training sets $\mathcal{T}_{\mathbf{A}}^{\mathrm{Train}}$, $\mathcal{T}_{\mathbf{B}}^{\mathrm{Train}}$ and testing sets $\mathcal{T}_{\mathbf{A}}^{\mathrm{Test}}$, $\mathcal{T}_{\mathbf{B}}^{\mathrm{Test}}$. Further, the set $\mathcal{T}_{\mathbf{A}}^{\mathrm{Train}}$ is further divided into two, not necessarily disjoint, subsets $\mathcal{T}_{\mathbf{A}}^{\mathrm{Cov}}$ and $\mathcal{T}_{\mathbf{A}}^{\mathrm{SVM}}$, used to estimate $\mathcal{C}_{v^{\mathbf{A}}}$ and the separating hypersurface respectively. We consider the performance of FINDER for balanced and unbalanced training sub-cohorts for the SVM estimation. In the Balanced regimes, $\mathcal{T}_{\mathbf{A}}^{\mathrm{SVM}}$ contains the first $N_{\mathbf{B}} - 1$ samples of $\mathcal{T}_{\mathbf{A}}^{\mathrm{Train}}$ and $\mathcal{T}_{\mathbf{A}}^{\mathrm{Cov}}$ contains the remaining $N_{\mathbf{A}} - N_{\mathbf{B}} - 1$ samples of $\mathcal{T}_{\mathbf{A}}^{\mathrm{Train}}$. In the Unbalanced regimes, $\mathcal{T}_{\mathbf{A}}^{\mathrm{SVM}} = \mathcal{T}_{\mathbf{A}}^{\mathrm{Cov}} = \mathcal{T}_{\mathbf{A}}^{\mathrm{Train}}$. Figure 4 illustrates the procedure for partitioning the data into training and testing subsets.

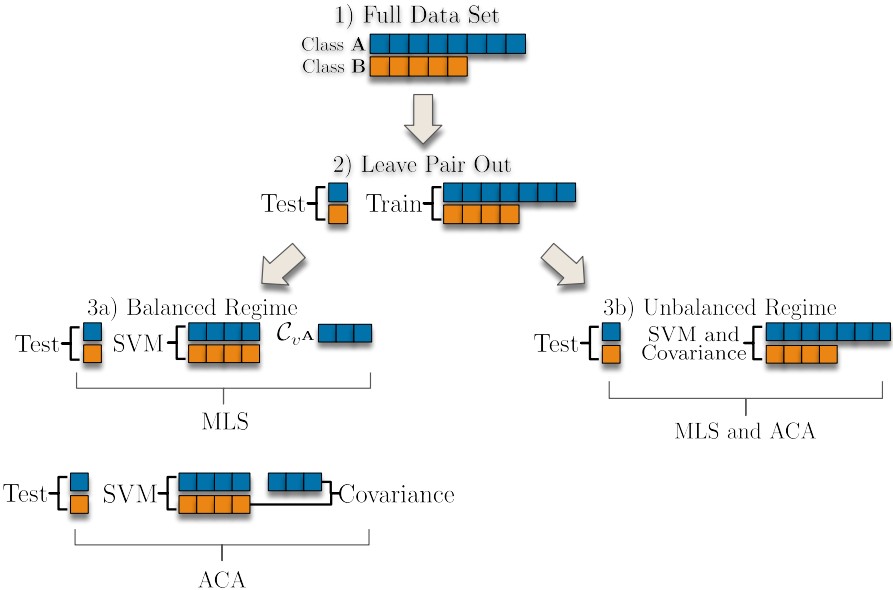

Figure 4: The division of the dataset $\mathcal{D}$ into training (which includes SVM separating hypersurface and covariance operator estimation) and validation (testing) subsets for one round of LPOCV.

In either regime, the aggregated dataset $\mathcal{T}_{\mathbf{A}}^{\mathrm{SVM}} \cup \mathcal{T}_{\mathbf{B}}^{\mathrm{Train}}$ is used to estimate the SVM.

We assume that all observations have had the empirical Class **A** expectation (computed as $\frac{1}{|\mathcal{T}_{\mathbf{A}}^{\mathrm{Cov}}|} \sum_{v_i^{\mathbf{A}} \in \mathcal{T}_{\mathbf{A}}^{\mathrm{Cov}}} v_i^{\mathbf{A}}$) subtracted.

For both FINDER methods, the covariance operator of $v^{\mathbf{A}}$ is estimated by

$$\widehat{\mathcal{C}}_{v^{\mathbf{A}}} := \frac{1}{|\mathcal{T}_{\mathbf{A}}^{\mathrm{Cov}}| - 1} \sum_{v_i^{\mathbf{A}} \in \mathcal{T}_{\mathbf{A}}^{\mathrm{Cov}}} v_i^{\mathbf{A}} \otimes v_i^{\mathbf{A}}$$

The eigen-pair estimates $\left\{\widehat{\lambda_i^{\mathbf{A}}}, \widehat{\phi_i^{\mathbf{A}}}\right\}_{i=1}^{|\mathcal{T}_{\mathbf{A}}^{\mathrm{Cov}}|}$ are computed as the eigen-pairs of $\widehat{\mathcal{C}}_{v^{\mathbf{A}}}$.

For ACA only, $\widehat{\mathcal{C}}_{v\mathbf{B}}$ is calculated as

$$\widehat{\mathcal{C}}_{v\mathbf{B}} = \frac{1}{|\mathcal{T}_{\mathbf{B}}^{\text{Train}}| - 1} \sum_{v_i^{\mathbf{B}} \in \mathcal{T}_{\mathbf{B}}^{\text{Train}}} (v_i^{\mathbf{B}} - \overline{v_i^{\mathbf{B}}}) \otimes (v_i^{\mathbf{B}} - \overline{v_i^{\mathbf{B}}})$$

where $\overline{v_i^{\mathbf{B}}} = \frac{1}{|\mathcal{T}_{\mathbf{B}}^{\text{Train}}|} \sum_{v_i^{\mathbf{B}} \in \mathcal{T}_{\mathbf{B}}^{\text{Train}}} v_i^{\mathbf{B}}$. Note that unlike $\mathcal{T}_{\mathbf{A}}^{\text{Train}}$, the entirety of $\mathcal{T}_{\mathbf{B}}^{\text{Train}}$ in both the Balanced and Unbalanced regimes within ACA, is used to compute $\widehat{\mathcal{C}}_{v\mathbf{B}}$ and the SVM.

For the chosen value of truncation parameter $M_{\mathbf{A}} < |\mathcal{T}_{\mathbf{A}}^{\text{Cov}}|$, the isometry $V_{M_{\mathbf{A}}}$ as described in section B.1 is estimated by the operator satisfying

$$\widehat{V}_{M_{\mathbf{A}}} f := \sum_{i \in I} f(i) \widehat{\psi}_i$$

where $\left\{ \widehat{\psi}_i \right\}_{i \in I}$ is an orthonormal basis for $\left( \text{Span} \left\{ \widehat{\phi}_i^{\mathbf{A}} \right\}_{i=1}^{M_{\mathbf{A}}} \right)^{\perp}$ created by adapting the algorithm detailed in Tausch and White (2003) as in MLS, or by a full SVD if $F$ is small enough that a full SVD is faster.

The eigenvectors $\left\{ \widehat{t}_m \right\}_{m=1}^{M_{\text{res}}}$ corresponding to the smallest (largest if ACA-L) $M_{\text{res}}$ eigenvalues of the operator $\widehat{V}_{M_{\mathbf{A}}}^* \widehat{\mathcal{C}}_{v\mathbf{B}} \widehat{V}_{M_{\mathbf{A}}}$ are computed. Finally, the training data is updated as

$$v_i^{\mathbf{A}} \to \sum_{m=1}^{M_{\text{res}}} \left\langle v_i^{\mathbf{A}}, \widehat{V}_{M_{\mathbf{A}}} \widehat{t}_m \right\rangle_{\mathcal{H}} \mathbf{e}_m, \qquad v_i^{\mathbf{B}} \to \sum_{m=1}^{M_{\text{res}}} \left\langle v_i^{\mathbf{B}}, \widehat{V}_{M_{\mathbf{A}}} \widehat{t}_m \right\rangle_{\mathcal{H}} \mathbf{e}_m$$

With $\{\mathbf{e}_m\}_{m=1}^{M_{\text{res}}}$ the standard basis of $\mathbb{R}^{M_{\text{res}}}$.

**Remark C.1.** *The leave-one-out cross validation approach is applied to both classes. If we have two classes $\mathbf{A}$ and $\mathbf{B}$ with number of samples $N_{\mathbf{A}}$ and $N_{\mathbf{B}}$, one sample is removed from each class as validation and the rest as training. All possible combinations are removed for each class. This leads to a total of $N_{\mathbf{A}} N_{\mathbf{B}}$ training-validation tests.*

## D    EXPERIMENTAL DETAILS

We conduct a set of experiments spanning various kinds of inherently noisy datasets, sourcing data from bio-medical and geo-sensing contexts. For each setting, we pick datasets with high and low noise contents respectively, so we can compare performances of conventional vs our methods and how they each scale with an increase in noise. We give a brief description of each dataset below:

### D.1    AD USING BLOOD PROTEINS

To assess the performance of the FINDER framework, we employed proteomic data from the Alzheimer's Disease Neuroimaging Initiative (ADNI), a longitudinal, multi-center observational study initiated in 2003 to facilitate the discovery and validation of biomarkers for Alzheimer's disease progression. The ADNI dataset comprises multiple study phases—namely ADNI1, ADNIGO, ADNI2, ADNI3, and ADNI4—with clinical and molecular data collected across time points Petersen et al. (2010).

Our analysis focused on the ADNI1 cohort, consisting of 209 subjects clinically diagnosed with Alzheimer's disease (AD), 742 with Late Mild Cognitive Impairment (LMCI), and 112 cognitively normal (CN) controls. Peripheral blood samples were collected at baseline (BL) and at the 12-month follow-up (M12). The experimental evaluation in this section is carried out using the plasma proteomics subset M12, comprising quantitative measurements for 146 protein biomarkers. This final cohort distribution includes 54 CN, 96 AD, and 346 LMCI participants. Note that the LMCI paritipants are actually a combination of LMCI and MCI.

| AUC | | | | | |
|---|---|---|---|---|---|
| Regime | SVM Linear | SVM w/ RBF | Logit Boost | BAG | RUS Boost |
| AD vs. CN | 0.754 | 0.561 | **0.798** | 0.743 | 0.763 |
| Time (ms) | 127.66 | **102.79** | 1173.42 | 1335.89 | 599.03 |
| AD vs. LMCI | 0.768 | 0.590 | **0.790** | 0.781 | 0.768 |
| Time (ms) | 356.61 | **246.40** | 2302.33 | 2092.64 | 897.66 |
| CN vs. LMCI | **0.910** | 0.637 | 0.814 | 0.778 | 0.776 |
| Time (ms) | 246.69 | **179.36** | 2075.48 | 1339.35 | 796.28 |
| Accuracy | | | | | |
| Regime | SVM Linear | SVM w/ RBF | Logit Boost | BAG | RUS Boost |
| AD vs. CN | 0.644 | 0.486 | **0.684** | 0.684 | 0.657 |
| Time (ms) | 127.66 | **102.79** | 1173.42 | 1335.89 | 599.03 |
| AD vs. LMCI | 0.570 | 0.497 | 0.667 | **0.735** | 0.581 |
| Time (ms) | 356.61 | **246.40** | 2302.33 | 2092.64 | 897.66 |
| CN vs. LMCI | 0.717 | 0.497 | 0.586 | **0.721** | 0.513 |
| Time (ms) | 246.69 | **179.36** | 2075.48 | 1339.35 | 796.28 |

Table 3: Benchmark machine performance for ADNI blood protein data (raw data features), along with elapsed time to complete one round of LPOCV. Maximum AUC and accuracy and minimum run time are emboldened in each row.

| Balanced | | | | | | |
|---|---|---|---|---|---|---|
| Regime SVM | MLS Linear | MLS RBF | ACA-S Linear | ACA-S RBF | ACA-L Linear | ACA-L RBF | Best Benchmark |
| AD vs. CN | 0.722 | **0.819** | 0.752 | 0.783 | 0.710 | 0.783 | 0.684 |
| Time (ms) | 116.1 | 130.9 | **68.8** | 86.7 | 74.0 | 78.4 | 1173.42 |
| AD vs. LMCI | 0.561 | 0.736 | **0.788** | 0.767 | 0.525 | 0.775 | 0.735 |
| Time (ms) | 212.1 | 287.0 | **81.8** | 109.4 | 84.9 | 100.1 | 2092.64 |
| CN vs. LMCI | 0.783 | 0.786 | **0.918** | 0.907 | 0.748 | 0.835 | 0.721 |
| Time (ms) | 164.7 | 231.3 | **74.8** | 92.5 | 79.7 | 89.1 | 1339.35 |
| Unbalanced | | | | | | |
| Regime SVM | MLS Linear | MLS RBF | ACA-S Linear | ACA-S RBF | ACA-L Linear | ACA-L RBF | Best Benchmark |
| AD vs. CN | 0.788 | **0.857** | 0.796 | 0.855 | 0.790 | 0.857 | 0.684 |
| Time (ms) | 112.1 | 129.5 | **76.2** | 93.0 | 77.0 | 85.6 | 1173.42 |
| AD vs. LMCI | 0.559 | 0.735 | 0.784 | **0.778** | 0.558 | 0.734 | 0.735 |
| Time (ms) | 215.2 | 291.1 | **115.3** | 179.1 | 121.8 | 178.1 | 2092.64 |
| CN vs. LMCI | 0.761 | 0.791 | **0.897** | 0.841 | 0.715 | 0.790 | 0.721 |
| Time (ms) | 164.9 | 239.9 | **91.4** | 135.9 | 94.6 | 146.2 | 1339.35 |

Table 4: Maximum accuracy Achieved ADNI data sets along with elapsed time to complete one round of LPOCV. We observe that the benchmark accuracy is ameliorated across the three cohorts within the ACA-S regime (although the MLS w/ RBF regime offers the optimum accuracy in the AD vs. CN cohort, regardless of balancing). Additionally, each of the FINDER regimes offer a significant reduction in run time compared to benchmark.

Data used in preparation of this article were obtained from the Alzheimer's Disease Neuroimaging Initiative (ADNI) database (adni.loni.usc.edu). As such, the investigators within the ADNI contributed to the design and implementation of ADNI and/or provided data but did not participate in analysis or writing of this report. A complete listing of ADNI investigators can be found at: http://adni.loni.usc.edu/wp-content/uploads/how_to_apply/ADNI_Acknowledgement_List.pdf.

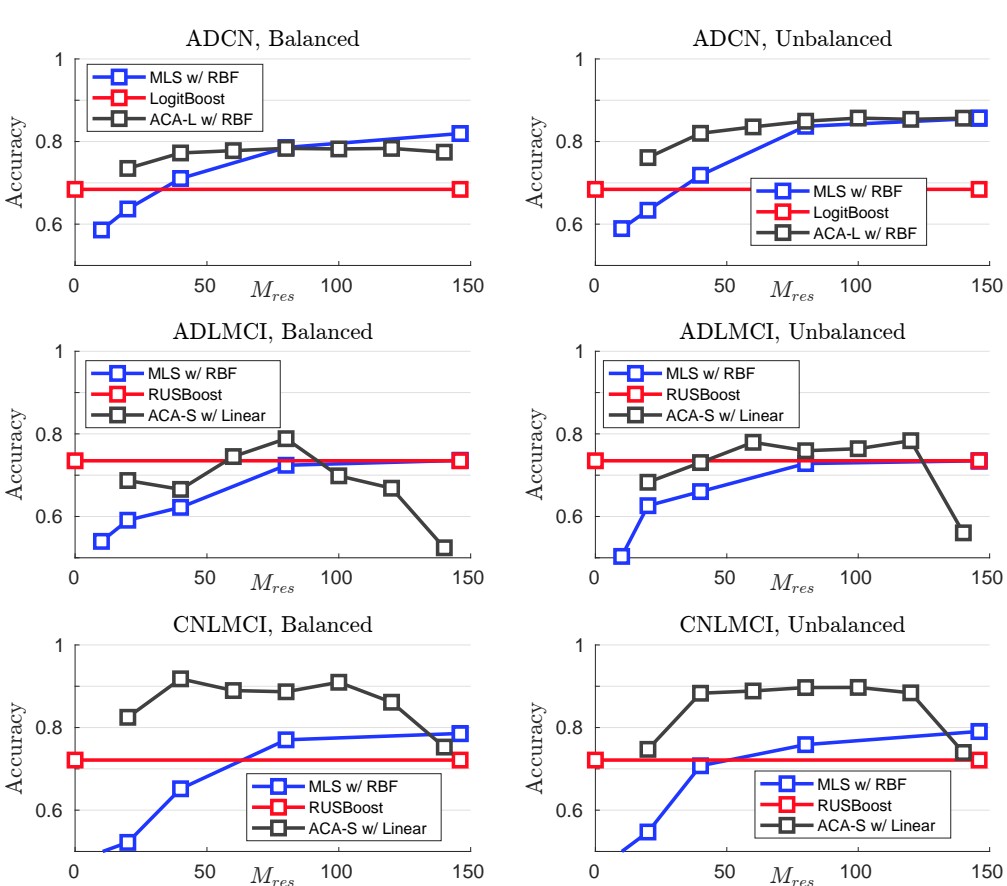

Figure 5: Accuracy obtained across all three methods for each of the three ADNI cohorts. Within each method (benchmark, MLS, ACA), we display the regime which obtains the maximum accuracy among the values of $M_{res}$ tested. We observe that the ACA-S regime either outperforms or matches both the MLS method and the benchmark learners across all three cohorts. However, the performance of the ACA-S regime can be sensitive to the choice of $M_{res}$, as exemplified in the AD vs. LMCI and CN vs. LMCI cohorts. Values of $M_{res}$ which are too high or too low can prevent the ACA-S regime from achieving maximum accuracy, and can even reduce the accuracy below benchmark level, as exhibited by the AD vs. LMCI cohort. The data demonstrate a significant improvement on the benchmark performance, prompting an ad-hoc analysis of the ratio of the FINDER and benchmark error rates as tabulated in Figure 6.

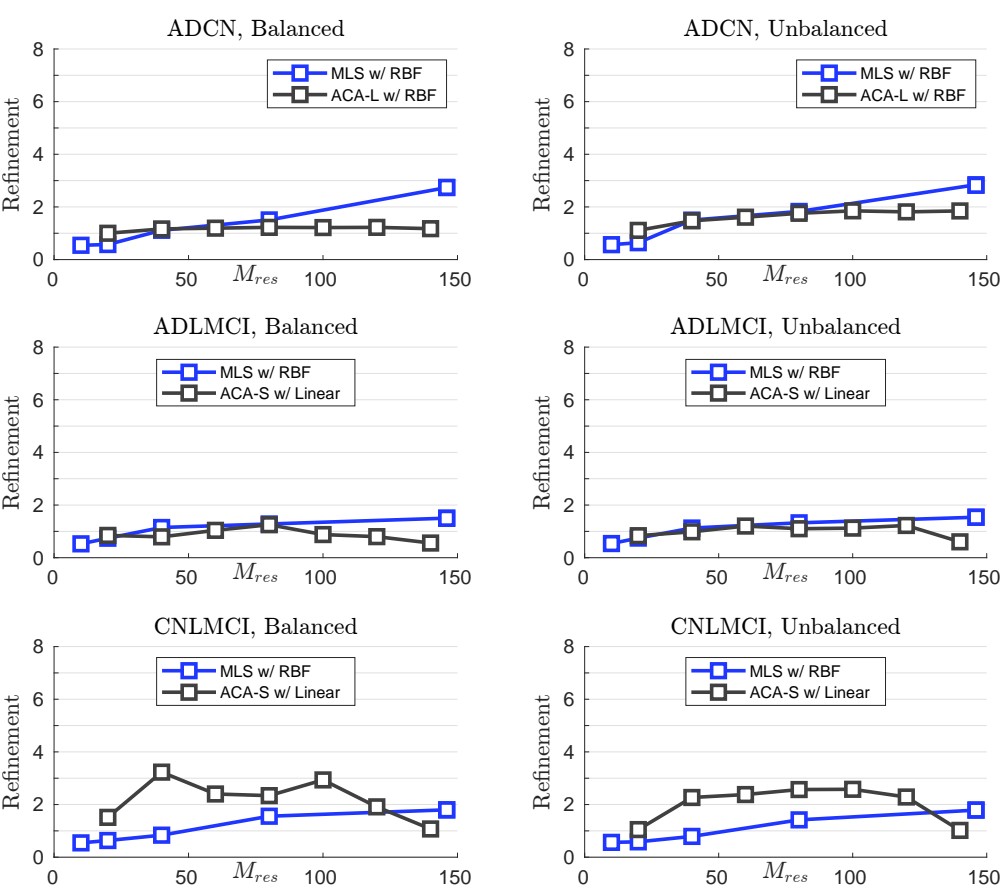

Figure 6: ADNI accuracy refinement with respect to best performing baseline, computed as $\frac{1 - \text{accuracy}_{\text{FINDER}}}{1 - \text{accuracy}_{\text{Benchmark}}}$. In both the Balanced and Unbalanced regimes, both the ACA and MLS methods offer a two fold (or near two fold) reduction in error across all three ADNI cohorts. In particular, the MLS method obtains an almost three fold error reduction in the AD vs. CN cohort, and the ACA method obtains an almost four fold error reduction in the CN vs. LMCI cohort.

## D.2 REMOTE SENSING (DEFORESTATION DETECTION)

The radar and optical data are provided by the Copernicus Sentinel data [2018-2022]'. From the optical Sentinel-2 the EVI data observations are split between training and validation. The training data consists of 71 optical Sentinel-2 EVI measurements between December 17, 2018 and March 21, 2020. Sentinel-2 covers the Earth every 5 days. However, some days contain heavy clouds and are thus removed from the dataset. The validation dataset consists of 161 days between March 26, 2020 to December 31, 2022. The SAR Sentinel-1 data consist of 234 samples between January 4th 2017 and December 28, 2022. In Figure 7, the detection capabilities of the FINDER approach with both optical and radar data are shown. This corresponds to a small region from the full testing region. Notice the regions where the forest is removed are detected with high accuracy. In particular, the road created by removing the trees in the yellow box is detected.

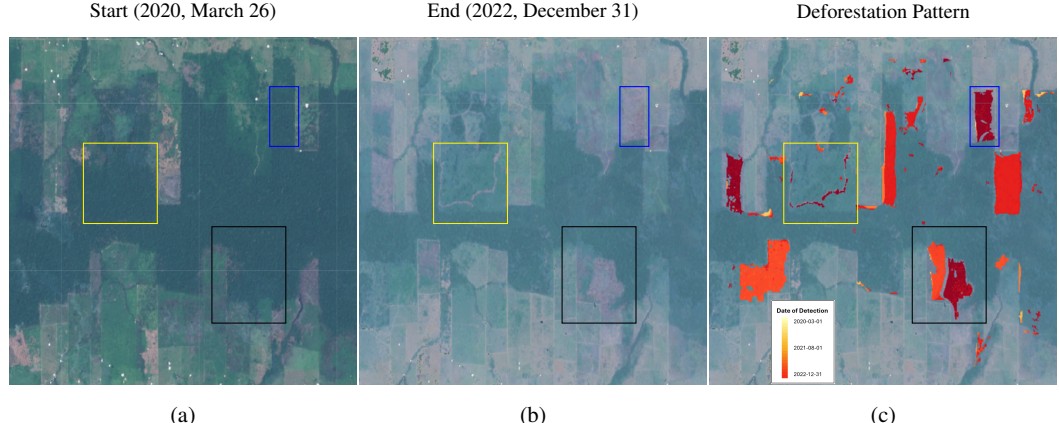

| Start (2020, March 26) | End (2022, December 31) | Deforestation Pattern |
|:---:|:---:|:---:|
| (a) | (b) | (c) |

Figure 7: Multi-sensor hybrid optical and SAR tracking of deforestation during a cloudy period. In these images we examine the deforestation pattern from March 26, 2020 to December 31, 2022 using the HMM model. (a) Due to the cloudy period, a composite image for the start is shown. (b) By the end date notice that many regions of the forest have been removed. (c) The color map shows the tracking of deforestation using the HMM with the optical and SAR data. There are many deforestation activities that are caught within the HMM hybrid model.

A natural question for the HMM + FINDER method is whether the anomaly filter is a necessary part of the pipeline. Table 5 compares the results from using the anomaly data against just using the raw optical data. Based on these values it would seem that using the raw optical data is better, however a closer inspection of the results illustrates a different picture. Figure 8 shows that while there is already good class separation in the unprocessed optical data between dry forest and dry bare ground, since water and bare ground both have low EVI values we get that wet forest and bare ground are classified together. This causes regions of false positives in marshy land, as can be seen in Figure 8 in the top right. Applying the KLE allows us to better separate these classes. Note there are still some false positives in these regions when using the processed optical data due to the radar data, which is also affected by water. While these regions of false positives are relatively small in this example, using the raw optical data could cause serious problems for regions experiencing heavy rainfall/flooding or with a large portion of marshy land - the anomaly data is still the superior choice.

Table 5: Accuracy using optical anomaly data (HMM + FINDER) vs unprocessed optical data (HMM)

| Algorithm (Data) | Training Days | Overall Acc. | User Acc. | Producer Acc. | Computational Time (h) |
|:---:|:---:|:---:|:---:|:---:|:---:|
| HMM + FINDER (Optical) | 71 | 0.931 | 0.823 | 0.718 | 13.95 |
| HMM + FINDER (Optical + Radar) | 71 | 0.942 | 0.878 | 0.745 | 49.47 |
| HMM (Raw Optical) | NA | 0.9668 | 0.8565 | 0.9105 | 9.82 |
| HMM (Raw Optical + Radar) | NA | 0.9614 | 0.8391 | 0.889 | 45.34 |

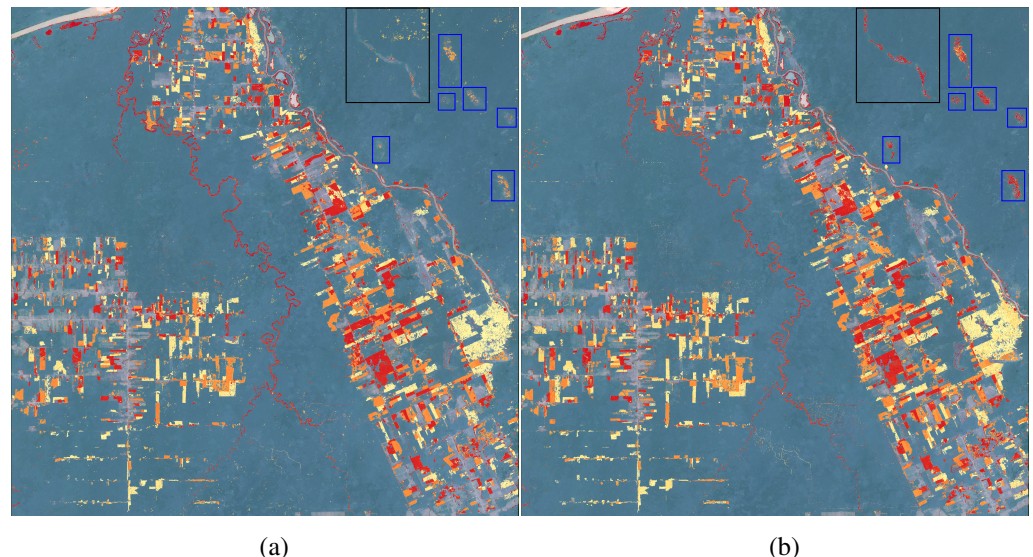

(a)                             (b)

Figure 8: (a) Hybrid datemap using anomaly values. (b) Hybrid datemap using unprocessed optical data. Application of the KL expansion results in better class separation between wet forest and deforested land, which both have low EVI values in the unprocessed optical data. This can be seen in the marshy land at the top right. Almost all of the detections in the black rectangles are removed, and the detections in the blue regions are reduced.

### D.3 ALZHEIMER'S DISEASE (AD) CLASSIFICATION FROM GENE EXPRESSION DATA (NEWAD)

The newAD dataset comprises the gene expression levels of 2053 enumerated genes from a cohort of 184 patients not afflicted with AD (Class **A**, Normal) and 145 patients afflicted with AD (Class **B**, AD). We put $U = \mathcal{H} = \mathcal{H}_M = \mathbb{R}^{2053}$ with $M_{\mathbf{A}} = 8$. This data was provided to us from Bhattacharya (2025). It is confidential at the moment and will not be made available to the public. Both the ACA and MLS methods are capable of elevating the AUC of binary prediction relative to the highest performing benchmark learner (SVM Linear), as detailed in Table 6 at the cost of a longer running time. At the same time, the ACA and MLS approaches also elevate the accuracy of binary prediction and reduce the run time compared to RUSBoost.

| Method | Score | Time (s) | Regime |
|--------|-------|----------|--------|
| AUC | | | |
| Benchmark | 0.761 | **0.51** | SVM Linear |
| MLS | **0.786** | 1.57 | Balanced, RBF |
| ACA-L | 0.785 | 1.21 | Balanced, RBF |
| Accuracy | | | |
| Benchmark | 0.697 | 6.84 | RUSBoost |
| MLS | 0.715 | 1.52 | Balanced, RBF |
| ACA-L | **0.725** | **0.76** | Balanced, RBF |

Table 6: Maximum AUC and accuracy achieved for newAD data by benchmark learners and both FINDER methods. Both metrics are improved marginally by the usage of FINDER. The FINDER methods more than double the run time to generate a comparable AUC with respect to benchmark, but significantly reduce the run time to generate a comparable accuracy with respect to benchmark.

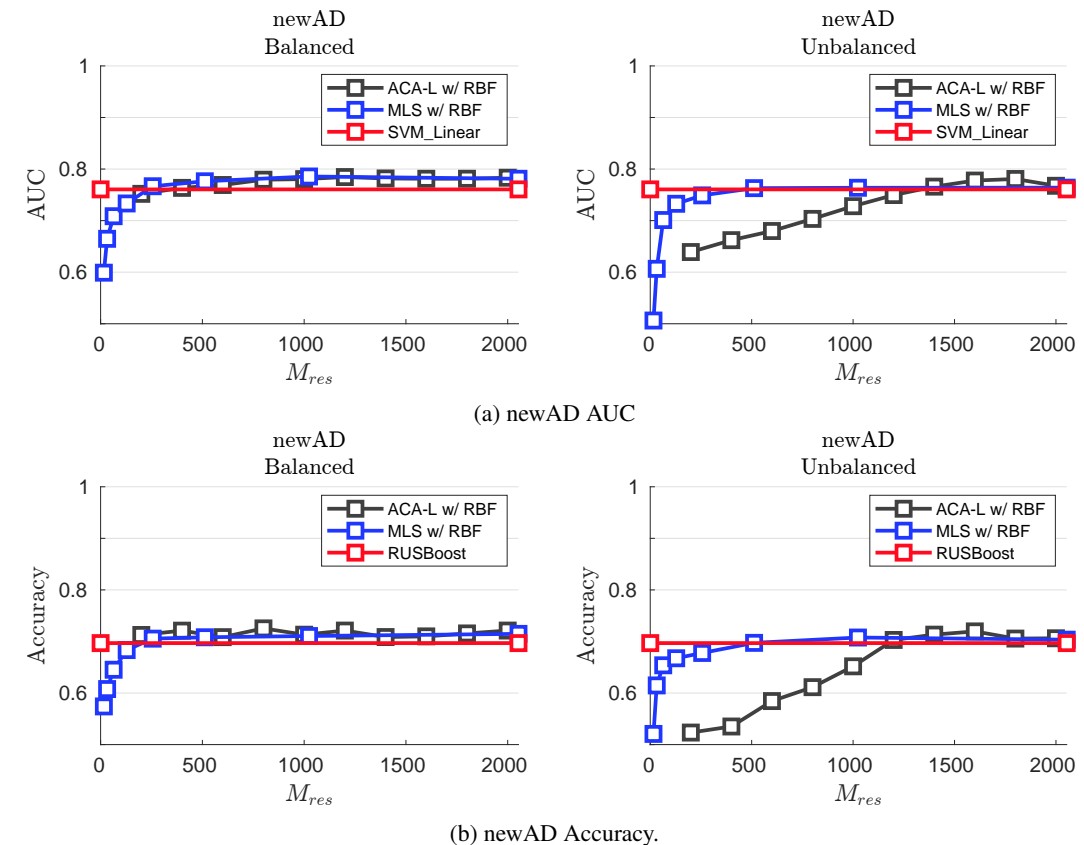

(a) newAD AUC

(b) newAD Accuracy.

Figure 9: The MLS method appears robust to the choice of pre-balancing versus not pre-balancing the data (if $M_{\text{res}}$ is sufficiently large). In contrast, ACA is highly sensitive to this choice, performing more consistently if the data is pre-balanced, although improvement upon the benchmark level is attained in both regimes. For both FINDER methods, this dataset tends to favor pre-balancing the data and the usage of an RBF separating boundary.

### D.4 CANCER CLASSIFICATION VIA GENE EXPRESSION DATA

The problem of cancer classification using gene expression datasets such as in Tan et al. (2005) is considered essentially solved: existing state of the art methods already perform at levels expected to be near optimal. Since it is a conventional example with an established corpus of results and analysis behind it, we will use it to sketch out how FINDER loses its relative advantages when features are aplenty or data is already well-behaved.

We consider the GCM gene expression dataset introduced by Ramaswamy et al. (2001). The data used in that paper is described in Tan et al. (2005). The dataset comprises gene expression profiles for $N_{\mathbf{A}} = 190$ tumor samples (Class $\mathbf{A}$) and $N_{\mathbf{B}} = 90$ normal tissue samples (Class $\mathbf{B}$), with each sample containing expression levels for $F = 16,063$ genes. The underlying domain $U$ is modeled as a one-dimensional interval $U := [0, F - 1]$, and gene expression signals are interpreted as discrete functions on $U$, represented by shifts of a Haar function $\chi$. We thus have $\mathcal{H} = L^2(U)$, $\mathcal{H}_M = \text{Span} \{\chi(x - k)\}_{k=0}^{16,062}$, and $\mathcal{H}_{\text{res}} = \mathcal{H}_{\mathbf{A}}^{\perp}$. The computation will be in the space $\mathbb{R}^{16,063}$.

Incorporating the multiscale features extracted by the FINDER method yields a marginal improvement in classification accuracy, as detailed in Table 7.

| Method | Score | Time (s) | Regime |
|--------|-------|----------|--------|
| AUC | | | |
| Benchmark | 0.966 | **1.81** | SVM Linear |
| MLS | 0.966 | 5.49 | Balanced, Linear |
| AC-LA | **0.969** | 12.37 | Unbalanced, Linear |
| Accuracy | | | |
| Benchmark | 0.931 | **1.83** | SVM Linear |
| MLS | **0.940** | 5.41 | Balanced, Linear |
| ACA-L | 0.934 | 12.34 | Unbalanced, Linear |

Table 7: Maximum AUC and accuracy achieved for GCM data by benchmark learners and both FINDER methods. The GCM data exhibit an inherent geometric separation such that an SVM with linear separating boundary can reliably segregate Class **A** from **B**. As such FINDER methods are capable of achieving a similar AUC and accuracy, but only offer a marginal improvement on both metrics at the cost of a significantly longer running time.

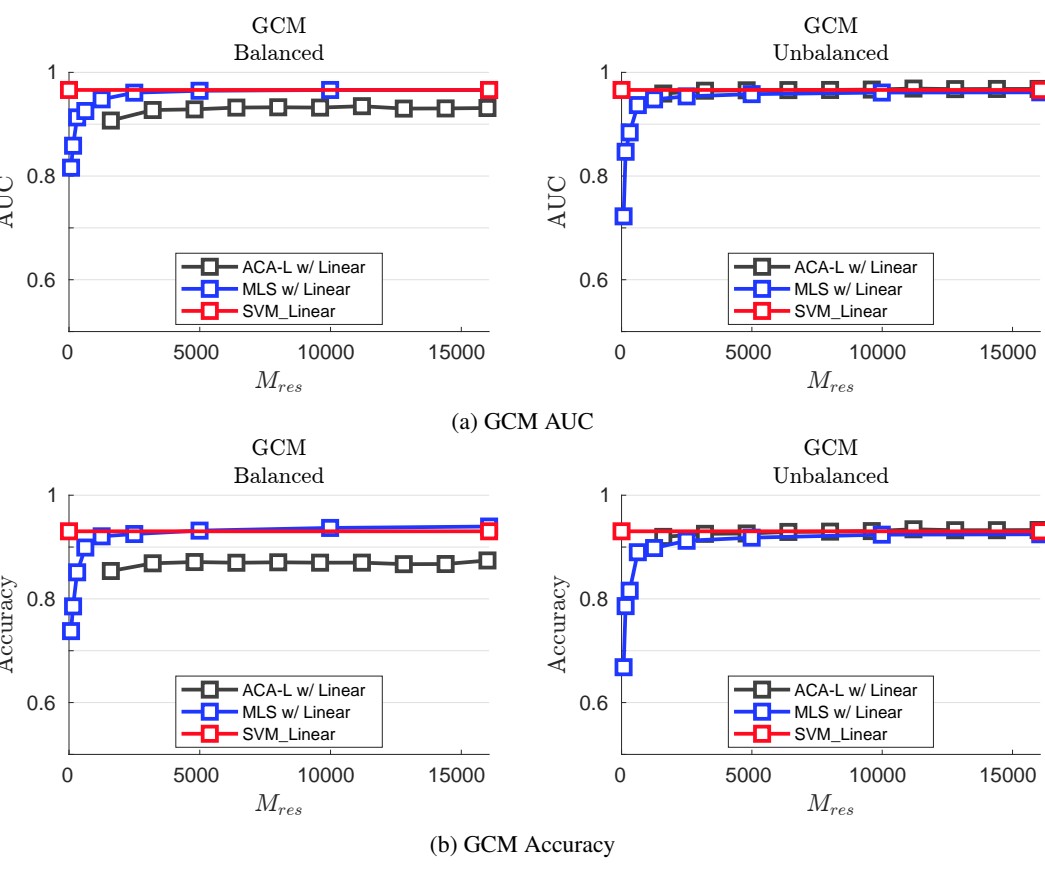

(a) GCM AUC

(b) GCM Accuracy

Figure 10: Best performing regimes among the three categories of learners for GCM data. The MLS method appears more robust the Balance vs. Unbalanced regimes as compared to the ACA method, in that the change in AUC and accuracy is not greatly affected by the choice to pre-balance the data or not. If the data is not pre-balanced, the ACA method performs at the benchmark level across several different residual subspace dimensions $M_{res}$, suggesting that this dataset is amenable to dimension reduction via the ACA-L regime.

## D.5 AD CLASSIFICATION FROM CSF DATA

The third and final AD-related dataset is the SOMAscan dataset, comprising a list of 7008 proteins obtained from the cerebrospinal fluid (CSF) of 167 AD patients (Class **A**) and 138 CN patients (Class

**B**). The raw data was imputed using a 5-nearest neighbors regression. We put $U = \mathcal{H} = \mathcal{H}_M = \mathbb{R}^{7008}$ with $M_{\mathbf{A}} = 8$. While the two FINDER methods fail to outperform LogitBoost on this dataset (see Table 8), they do provide an advantage vis a vis cost efficiency. Note that as expected, FINDER does augment results achieved by the conventional SVM methods, allowing them to reach AUC and accuracy levels that are closer to LogitBoost in less than half the time.

The AUC obtained by LogitBoost exceeds 0.95. For datasets with this level of separability, any noise typically comes from the collection and manual classification of the data itself, rather than any inherent variability in the underlying features. As such, the separability of this particular dataset may not be amenable to improvement by our FINDER methods. This is in contrast to datasets, such as the ADNI data cohorts, in which the baseline AUC/accuracy is well below the desirable 0.90 level.

| Method | Score | Time (s) | Regime |
|---|---|---|---|
| AUC | | | |
| Benchmark | **0.957** | 11.75 | LogitBoost |
| | 0.922 | 2.122 | SVM w/ Linear |
| | 0.899 | **1.927** | SVM w/ RBF |
| MLS | 0.930 | 4.51 | Balanced, RBF |
| ACA-S | 0.935 | 4.46 | Balanced, Linear |
| Accuracy | | | |
| Benchmark | **0.894** | 11.75 | LogitBoost |
| | 0.847 | 2.122 | SVM w/ Linear |
| | 0.819 | **1.927** | SVM w/ RBF |
| MLS | 0.862 | 3.34 | Unbalanced, RBF |
| ACA-S | 0.878 | 5.16 | Balanced, RBF |

Table 8: Maximum AUC and accuracy achieved for CSF data by benchmark learners and both FINDER methods. The FINDER methods more than halve the run time of Logitboost at the cost of a marginally lower AUC and accuracy. At the same time, the FINDER methods elevate the AUC and accuracy obtained by the SVM (with linear and RBF separating hypersurface) with the caveat of a longer run time.

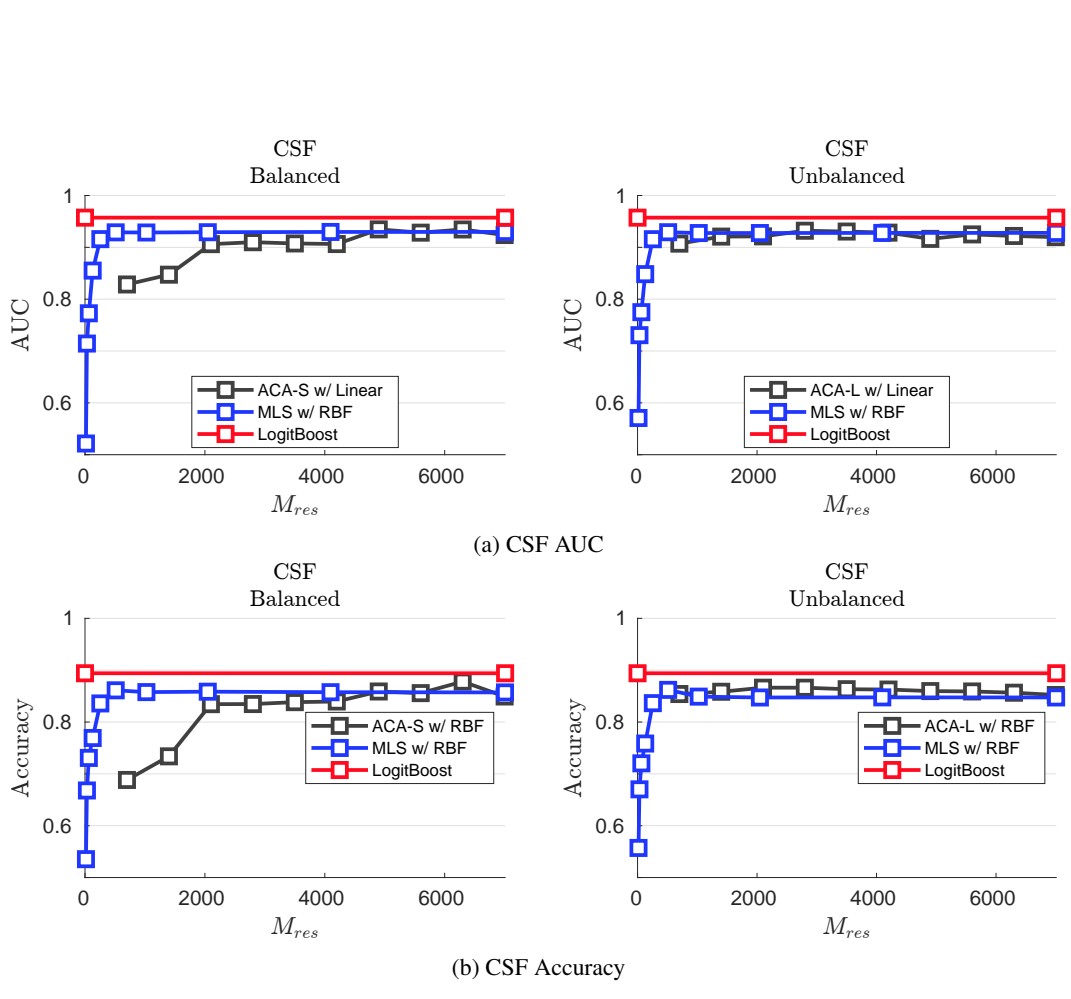

(a) CSF AUC

(b) CSF Accuracy

Figure 11: Both FINDER methods achieve a lower accuracy and AUC compared to the best benchmark (LogitBoost). The performance of MLS remains robust to pre-balancing versus not pre-balancing the data, while the ACA method is more sensitive to this choice. Within the Unbalanced regime, the ACA achieves a more consistent AUC and accuracy across different values of $M_{res}$, suggesting that this method is better suited to dimension reduction. However, the highest AUC and accuracy is achieved within the ACA-S, Balanced regime

