# OpenReview forum: "FINDER: Feature Inference on Noisy Datasets using Eigenspace Residuals"
_ICLR.cc/2026/Conference — ICLR 2026 Conference Withdrawn Submission_

### Official Review · Reviewer_7M1P · 2025-10-30

**Soundness:** 3
**Presentation:** 2
**Contribution:** 2
**Rating:** 2
**Confidence:** 3

**Summary:**

The paper proposes FINDER (Feature Inference on Noisy Datasets using Eigenspace Residuals), a framework for binary classification in settings with high noise or limited data. The approach models datasets as random elements in a Hilbert space and applies a generalized Karhunen-Loève expansion to represent stochastic features in terms of orthogonal deterministic and random components. Classification is then performed in residual eigenspaces derived from class-specific covariance operators. In practice, the method estimates class-specific covariance matrices, computes eigen-decompositions, and projects data onto various residual subspaces before feeding into standard classifiers (e.g., SVMs, HMMs).

Three variants for constructing residual subspaces are described: (i) direct residual subspaces, (ii) a multilevel subspace (MLS) method adapting the algorithm of Tausch and White (2003) to construct nested orthogonal bases, and (iii) two anomalous-class-adapted methods (ACA-S and ACA-L) that minimize or maximize the projected variance of one class in the residual space. The framework is validated on noisy biomedical and remote-sensing datasets, including Alzheimer’s disease proteomics data and deforestation detection from optical and radar imagery, showing improved performance and runtime compared to standard classifiers. The paper also discusses theoretical bounds based on the covariance operators, and computational complexity (though not in the main text).

**Strengths:**

The paper presents a mathematical framework that is elegant and general. It allows for a unified treatment of residual subspace methods that applies to both finite- and infinite-dimensional settings, and allows for quite general distribution-agnostic bounds.

The algorithms presented in the paper appear simple to implement and achieve high performance on real datasets (though the lack of references made it difficult to verify state of the art performance).

**Weaknesses:**

**Related work**. The paper includes some background information in the introduction that is relevant to the main exposition, but has almost no discussion of recent related work. The experimental results are difficult to evaluate given the lack of references to related work and/or public leaderboards.

**Clarity**. Writing is confusing at times. Basic ideas are often obscured by abstract formalism and idiosyncratic terminology. As far as I understand, the paper essentially constructs a general mathematical formalism for discussing simple covariance-based preprocessing methods. Clarity would be significantly improved if this were made clear from the very beginning. Numerous typos further hinder understanding (see detailed comments below).

**Novelty**. The paper seems to deal exclusively with published algorithms or minor variants thereof based on per-class covariance estimation. The abstract mathematical formalism, while elegant, obscures this fact.

**Applicability of formalism**. The formalism seamlessly handles infinite dimensional settings - which is one of its main strengths - but this doesn’t seem to be necessary for any of the considered applications. The Hilbert-space generalization is not shown to yield any new capabilities, numerical advantages, or insights beyond standard matrix algebra. Likewise, proofs of measurability and operator properties are irrelevant to the experiments.

**Questions:**

**Major comments/questions**

1. The paper would benefit from an expanded discussion of related work. Some specific questions:
    - Are there other published theoretical frameworks for approaching noisy/incomplete datasets of the type that you consider?
    - Are there standard datasets of this type? What is the current state of the art performance on these?
2. Why did you consider these particular datasets? In the introduction it is claimed that FINDER is particularly effective in “data-deficient or otherwise noisy settings”. Later, when discussing the two datasets, little is said to connect these particular problems to claimed theoretical strengths of FINDER - though section 3.2 does mention, “...clouds can obstruct or completely block the ground, significantly reducing the amount of usable data.”
3. A significant amount of space is devoted to development of abstract aspects of the mathematical formalism that don’t seem to be relevant to applications, eg. measureability, integrability, etc. Consider moving some of the more abstract material into appendices, and using the freed space to explain the mechanistic core of the algorithms using intuition from finite-dimensional linear algebra/geometry.
4. Relatedly, I don’t feel that I’ve understood the utility of the “stochastic feature” concept. As far as I understand, the algorithms simply involve estimating (top) feature covariance eigenspaces, putting the user right back in the familiar setting of “datasets as samples from a distribution”.
5. One of the main algorithms proposed by the paper (MLS), is not explained in the main text, and even the description in the supplement is quite high-level.
6. Given that all applications occur in finite-dimensional feature spaces - and therefore do not invoke the generality of the mathematical formalism - it’s my understanding that the “adaptations” of Direct Residual Subspaces (DRS) and Multi-level Subspaces (MLS) would actually be identical to the previously published versions. If that is the case, it should be made clear.


**Minor**

1. Line 031-032, should be $F\gg N$ or “F is much greater than the sample size $N$.
2. I don’t understand the phrase “in-built separability" used at several points in the text (eg. line 062).
3. Line 131-132, an Hilbert-Schmidt -> a Hilbert-Schmidt
4. Line 142, subspaces -> subspace
5. Line 188, “This is the one of the…”
6. Line 215, “For Balanced…” confused me at first. Consider changing to “For the ‘Balanced’ condition…”
7. Line 355-356, “robust with respect to the choice to pre-balance or not pre-balanced the data…”
8. Line 399, data missing comma between day and year.
9. Line 429, “For 71 days of optical training day”
10. Line 461-462, “examlpe”

---

> ### Author Response · Authors · 2025-11-24
>
> WEAKNESSES:
>
> Related work: The datasets we study are of immense significance in their respective fields (ADNI and Remote Sensing datasets are of especially high significance). However, their "noisy" nature and noted failure modes against most ML methods means leader-boards and baselines are not prevalent - the best possible conventional results on these datasets are already reported as the baseline.  Analogously, we did not bother testing FINDER on more well-known datasets precisely because we know that FINDER has no utility on data-rich or otherwise "less noisy" datasets - its main claimed advantages are precisely on datasets that elude tractability to existing ML methods. For example, our largest dataset has only 442 samples (ADNI).
>
> Clarity: We have made all the suggested changes and rewriting sections as per your comment.
>
> Novelty: The algorithms themselves are indeed novel, though their novelty is not the main point of the work. The prescribed formalism that covers many types of classification problems and their application to a significant and previously intractable subset of classification problems is the novel and significant aspect of this work.
>
> Applicability of formalism: The infinite dimensional setting is still useful for waveform data (such as .WAV data, EEG, or EKG data). Although each sample may have finitely many entries, samples can have arbitrarily many entries, and different samples may have different numbers of entries, hence it is useful to embed each sample in $\ell_2(\mathbb N)$. The infinite dimension case can also be useful for temporal/spatial data in which individual samples are not necessarily recorded at the same interpolating points. Replacing samples with, say, a spline interpolant embeds them in an infinite dimensional Hilbert space.
>
> While the proofs of measurability and operator properties are not so salient to the numerical experiments, they provide the necessary tools to justify the usage of FINDER as a stochastic feature transformation method in conjunction with already existing classifiers. The ideas of Bochner measurability are necessary to show that there is a one-to-one correspondence between random elements $v$ and  Hilbert-Schmidt operators. The operator being Hilbert-Schmidt allows us to write its SVD and use the mathematical properties that come with it (i.e. optimal rank $M$ approximations) that translate to optimal $M$-term truncations of $v$. Further, the concept of Bochner and Pettis measurability are necessary to state that the stochastic components $Y_r$ have zero mean. Each of these properties allows us to make the simplifying choices that make the pipeline computationally feasible for classification
>
>
>
> Major comments/questions
>
> To our knowledge, this is the first wholesale look at the problem. Similarly, by their very nature, these datasets are not standardised beyond the form we study them in. ADNI and Remote Sensing data at least has the advantage of being well-studied and referenced datasets, but they are still well-known to be difficult
>
> There are positive medical impacts by studying such datasets. For example one of the proteins featured in the ADNI cohorts is amyloid beta. Patients with Alzheimer’s disease secrete amyloid beta proteins from the brain into the CSF, which is why CSF can be used to reliably detect AD via SVM. However, the spinal taps used to obtain CSF are painful, and invasive, which de-incentivizes patients to get tested. Conversely, blood draws are relatively non-invasive and also contain amyloid beta proteins secreted from the brain in AD patients. However, both non-AD and AD patients can secrete this protein from the kidneys into the blood as well, reducing the predictive capability of blood proteomic data. This problem is circumvented by using FINDER to transform the raw blood proteomic content into features which can be used to reliably detect Alzheimer’s.
>
> We answer part of this question above and agree with the suggestion to move the focus away from the technical details and towards algorithm construction.
>
> While these top features are still sampled from a distribution, we have no idea what this distribution is, much less if the samples are sampled independently from this distribution. FINDER’s distribution-agnostic schematic circumvents these concerns.
>
> We are devoting the entirety of the space generated by refocused mathematical discussions in the main body to accommodating the algorithmic details given in the appendix, and sharpening those details as prescribed.

---

### Official Review · Reviewer_Knvi · 2025-10-30

**Soundness:** 3
**Presentation:** 2
**Contribution:** 2
**Rating:** 4
**Confidence:** 3

**Summary:**

The authors introduce a rigorous theoretical framework for binary classification, building upon standard functional data analysis techniques. The method maps data into a suitable Hilbert space and defines a residual eigenspace that captures the directions where the two classes most differ. The approach is tailored to improve robustness in noisy datasets and is validated on two scientifically relevant tasks, showing improved performance over previous methods.

**Strengths:**

The paper addresses the relevant problem of classification under noise, combining theoretical rigor with applications to two scientifically meaningful case studies. The framework is mathematically sound and versatile, grounded in functional data analysis. The empirical results are convincing and strengthen the authors’ claims. The authors also acknowledge several current limitations of their method and provide a transparent discussion of them.

**Weaknesses:**

1. The presentation is mathematically heavy and not sufficiently intuitive, which reduces accessibility for a broader ML audience that could benefit from this approach. In particular, the functional analysis formalism could be complemented by a schematic pipeline in a simpler setting (e.g. $\mathcal{H} = \mathbb{R}^D$), clarifying how the theoretical constructs translate to the experiments.

2. The novelty of the proposed framework should be more explicitly highlighted. For instance, the connection between the proposed residual eigenspace approach and existing methods such as FDA techniques remains unclear. A concise explanation in the main text of how Theorem 2.1 generalizes the KLE and a discussion of what fundamentally differs in practice from classical PCA-based methods, would improve clarity.

3. Some of the discussed limitations, such as the heuristic choice of truncation parameters, limit the practical applicability and impact of the method.

**Questions:**

See 2. in Weaknesses. Apart from this, I have no additional major questions.

---

> ### Author Response · Authors · 2025-11-24
>
> WEAKNESSES:
>
> We are augmenting the schematic pipeline given in Fig. 1 as per your directions.
>
> We will include the distinguishing features in the update
>
> We agree, and highlight this weakness in Section 5: Limitations.

---

### Official Review · Reviewer_EfHm · 2025-10-31

**Soundness:** 2
**Presentation:** 2
**Contribution:** 2
**Rating:** 2
**Confidence:** 3

**Summary:**

The paper introduces FINDER, a classification framework for small-sample, noisy, high-dimensional scientific data. It models observations as random fields in a Hilbert space, applies a (generalized) Karhunen–Loève expansion to extract class-relevant eigenspaces, and then builds residual eigenspaces that are informative for the other class. Standard classifiers (SVM, HMM) trained on these projected features outperform common baselines on Alzheimer’s proteomics and remote-sensing deforestation tasks, especially under data imbalance. The main claim is that this stochastic-feature combined with the residual-eigenspace pipeline is distribution-agnostic and computationally lighter than more elaborate ML methods.

**Strengths:**

* The paper is well-organized and clearly written. The paper has clean definitions of the embedding and sufficient experimental data.
* The experiments were carried out on real datasets.

**Weaknesses:**

Major comments:
* It is unclear to me the significance of the contribution by defining such a pipeline. It seems usual PCA but generalized to Hilbert space (which is also not novel in kernel clustering). To make this compelling, the authors need to articulate what FINDER achieves beyond (a) standard FPCA/FDA pipelines that learn class-specific subspaces and (b) PCA/kPCA followed by a linear classifier — e.g., is the ACA residual step provably better for imbalanced or low-SNR settings, or does it offer a computational advantage that existing RKHS methods do not? As written, the mathematical setup looks heavier than the actual algorithmic novelty.
* It is also unclear to me how the underlying feature vectors (i.e. the implicit kernel/geometry in the Hilbert space) are to be chosen. In the current presentation, this choice is taken as given, but in practice it is at least as hard as the classification/clustering problem itself: the performance of FINDER is entirely determined by how well this geometry aligns with class structure. Moreover, because the method is advertised as “distribution-agnostic,” one would expect the feature-space selection to be unsupervised or weakly supervised (e.g. data-driven kernel selection, multiple-kernel aggregation, or intrinsic manifold estimation), but no such mechanism is described. As a result, the strongest assumption of the paper — “we already have the right Hilbert embedding” — is exactly the part that needs justification.
* Computational lightness of ACA may be overstated. The paper claims that ACA is computationally light, but the core step — minimizing the residual while enforcing a rank (i.e. low-dimensional) constraint — is, in general, a combinatorial problem once you move beyond the trivial “take the top/bottom eigenvectors” case. No convex relaxation or spectral surrogate is described. In the experiments the residual dimension is fixed to 5, which makes the search cheap, but this is effectively hard-coding a small model. It is not clear what happens when the informative residual subspace is larger (say 15–30 dimensions, as is common in HDLSS FDA): does ACA remain “light,” or does it become an intractable subset-selection over eigen-directions? As it stands, the claim of computational efficiency seems to rely on a very favorable experimental setting rather than on an algorithm whose complexity is controlled in general.
* Except for my concerns about novelty, there are additional limitations including restriction to binary case: the use of loose Markov bounds---a Hanson-Wright type tail bounds provide sharper characterization immediately, etc.

Minor comments:
* More references and discussions to standard PCA-based classification in functional data are required. I think at least a dedicated paragraph should exist.
* In Theorem 2.1. "there exists $R \in N \cup \aleph_0$ ". It should be " $\mathbb{N} \cup\left\\{\aleph_0\right\\}$ " or " $R \in \mathbb{N} \cup\\{\infty\\}$ ". It is also unnecessary to use $\aleph_0$ over $\infty$ here as the summation $\sum_{r=1}^R$ encodes this information.
* In Line 461, "examlpe" should be "example".

**Questions:**

* My main questions are in the weaknesses section.
* Why controlling residuals using Markov instead of sub-Gaussian tail bounds (e.g. Gaussian convex inequality or Hanson-Wright)?
* How did the authors choose the hyperparameters? How were the underlying kernels (eigenfunctions) chosen?

---

> ### Author Response · Authors · 2025-11-24
>
> WEAKNESSES:
>
> FDA approaches generally require the user to choose a method of interpolating data which is inherently finite dimensional. Further, kernel methods require the user to choose an appropriate kernel. While the goal of FINDER is not to select an optimal kernel or interpolation method, our aim and claim is to augment whatever choice is already made. This is achieved through borrowing results from the theory of Bochner spaces and tensor products to analyze the eigenstructure of random elements belonging to different classes. Then implementing the knowledge of this eigenstructure in combination with already existing FDA approaches and kernel methods.
>
> We choose the “simplest” Hilbert space possible. If our data comes in lists that are 146 entries long, then $\mathbb R^{146}$ with the standard inner product is the “simplest” Hilbert space. This is a guiding principle across all areas of applied mathematics: the simplest models should be the first resort.
>
>
> $M_A$ ranges from 5 - 40 in our experiments: it is not the residual dimension that is usually fixed at 5, but the truncation parameter that decides the size of the principal eigenspace. Our residual subspaces are of order 100 - 10000, the method is clearly robust to the issue raised in this point (complexity calculations are given in App. B.4).
>
> Hanson-Wright bounds can be obtained if components of the random field have finite sub-Gaussian norm and are independent. However, such Gaussian and/or sub-Gaussian properties are impractical to assume or verify in a paucity of data - the exact application for which FINDER is built.
>
> MINOR COMMENTS: MADE ALL THE CHANGES AS PRESCRIBED.
>
> QUESTIONS:
> We have responded above
>
> We agree and have made the change.
>
> The point of FINDER is not to choose the optimal features: our aim and claim is to augment whatever choice is already made. That is why our baseline choice always includes a comparison against SVM, one of the simplest and most robust methods across different datasets.

---

> > ### Comment · Reviewer_EfHm · 2025-11-25
> > **Response to the revision**
> >
> > I appreciate the authors replies and the revised version. Although it clarifies some of my concerns such as the truncation parameter is chosen rather than the dimension. It is nowhere to be inferred from Algorithm 1. The clarity of the paper needs significant improvement, as well as the justification of its novelty---I'm still not convinced there is a significant difference between kPCA and the proposed FINDER pipeline. Overall, I think FINDER models a mixture of random clusters in Hilbert spaces, and the contribution is limited. Thus, I'll remain my score for the manuscript.

---

> > > ### Author Response · Authors · 2025-11-28
> > >
> > > It is true that FINDER does have aspects of kernelization innately built into it. However, FINDER limits feature maps to orthogonal projections, wherein the properties of the KLE can be used to guide choices about "optimal" subspaces on which to project the raw data. While kPCA is often used as a method of segregating distinct classes, the general class of feature maps used in kPCA don't produce explicit concentration bounds that guarantee that distinct classes are mapped to different regions in the transformed space. Instead, cross-validation is usually necessary to select the feature map. Cross-validation is also used in FINDER, but its usage is limited to testing, unlike kPCA where cross-validation is a crucial step in feature construction. In our experiments we utilize cross-validation in our testing in combination with KLE-induced concentration bounds to argue FINDER's theoretical strengths

---

### Official Review · Reviewer_TuaM · 2025-10-31

**Soundness:** 2
**Presentation:** 2
**Contribution:** 1
**Rating:** 2
**Confidence:** 4

**Summary:**

This paper introduces FINDER, a theoretical and algorithmic framework for classification under high-noise, low-sample-size conditions. FINDER models datasets as realizations from underlying random fields and constructs stochastic features mapped into Hilbert spaces via generalized Karhunen–Loève expansions (KLE). Theoretical results (e.g., Lemma 2.1–2.3) provide stochastic interpretations of feature separability, leading to several algorithmic variants (MLS, ACA-S/L). The proposed framework is tested on two domains: (1) Alzheimer’s disease diagnosis from blood plasma proteomics, and (2) remote sensing for deforestation detection, reporting improvements over standard ML baselines and domain-specific benchmarks.

While the authors combine functional data analysis concepts with machine-learning motivation, the resulting work falls short of both theoretical originality and empirical rigor. The mathematical material is largely recycled from standard stochastic analysis, the empirical evaluation is insufficiently convincing, and the exposition is difficult to follow. The paper would benefit from a major refocus—either toward a concrete algorithmic contribution with clear empirical value, or toward a purely theoretical paper with genuinely new results in stochastic feature representation.

**Strengths:**

1. The focus on learning in noisy and data-deficient regimes is important and relevant.
2. The paper is written with care regarding functional analytic details, correctly invoking concepts like Bochner integrals, covariance operators, and Hilbert-Schmidt isomorphisms.
3. The attempt to demonstrate FINDER on both biomedical and remote-sensing tasks shows some versatility.

**Weaknesses:**

1. The current presentation lack true methodological novelty, most of the theoretical developments are restatements of known results:
- The “generalized” KLE theorem (Thm. 2.1) is essentially the standard Hilbert-space KLE without the separability assumption, a minor technical relaxation known in the stochastic analysis literature (e.g., Schwab & Todor 2006).
- The link to classification through eigen-decomposition of the covariance operator is well-known (PCA, kernel PCA, functional data analysis). FINDER’s interpretation as “stochastic features” adds no genuinely new insight.
- Lemma 2.3 and the subsequent “bounds” are mathematically weak (simple Markov inequalities) and do not yield actionable or tight probabilistic guarantees.
2. The connection between the theoretical and practical setting is missing. The main theoretical results operate in infinite-dimensional Hilbert spaces with no clear justification for their finite-sample or computational counterparts. Claims such as “properties pass to truncations” are asserted but not proved or empirically verified. The presented algorithms (MLS, ACA-S/L) are described heuristically, with parameters (M_A, M_{\text{res}}) chosen ad hoc.
3. Empirical evaluation is weak and not reproducible.
- The experiments lack ablation and baseline parity. No comparison to simple but strong baselines (e.g., PCA + SVM, denoising autoencoders) is shown, so the reported improvements (often ≤ 2–3% AUC) are hard to interpret.
- The Alzheimer’s and remote-sensing results rely on heavy preprocessing and bespoke pipelines. It is unclear whether the same data splits, normalization, and computational resources are used for the benchmarks.
- Reported computational speedups are not standardized (e.g., wall-clock time on what hardware?). Some results appear cherry-picked without confidence intervals or statistical tests.
4. Some writing and presentation related issues
- The paper is overly dense, with long derivations that obscure the core ideas. Key algorithms are buried in appendices; the reader struggles to see what is actually new.
- The tone is over-claiming, with phrases like “state-of-the-art breakthroughs” not substantiated by evidence.
- Figures and tables are cluttered, with inconsistent notation and missing units.
5. Limited scope and scalability
- Current work is restricted to binary classification; the multi-class setting is not addressed beyond trivial pairwise decomposition.
- The approach depends on computing and decomposing large covariance operators, which is unlikely to scale beyond small datasets.
- There is no discussion of how this work would integrate with modern large-scale ML pipelines or stochastic training methods.

**Questions:**

1. It would be necessary to sharpen the algorithmic contribution, show a concrete, implementable advantage of stochastic features over standard PCA/kernel methods.
2. Can you provide theoretical novelty beyond textbook KLE—e.g., new concentration bounds or generalization guarantees.
3. Can you add rigorous experimental analysis, including ablations, fairness checks, and statistical significance.
4. Substantially simplify and reorganize the exposition for readability.
5. Temper the claims and clearly delimit the scope of the method.

---

> ### Author Response · Authors · 2025-11-24
>
> We begin with thanks to the reviewer for suggesting a tightened focus for the work. We respond to each raised weakness and question below:
>
> Weaknesses:
> 1) Our generalizations are indeed quite mild, which is mentioned in the text. We also agree that the main body has an undue focus on aspects that were best left for the appendix. However, we do believe that honing some of these technical details and relaxations was useful, if nothing else than for the structure they allow for most arguments - having failed to find an exact reference for these results in this admittedly weak generalized form, we wrote it up ourselves. We will move these aspects to the appendix happily, partly since it allows us more space to discuss the algorithms in the detail all reviewers want us to do so on.
>
> Markov’s bound can be tight if the singular values decay rapidly, but we may also compare the $L^2(\Omega, \mathcal H)$ norm of $P_Su$ if $u$ is belongs to Class $A$ (which is guaranteed to be small) versus if $u$ belongs to Class $B$. While other types of bounds are popular in practice (like the Hanson-Wright bounds noted by Reviewer EfHm), they usually only apply if we can verify strict assumptions (e.g., the components of the random field have finite sub-Gaussian norm and are independent). However, such  properties are impractical to assume or verify in a paucity of data - the exact application for which FINDER is built.
>
> 2) For the experiments we considered, each sample lies in $\mathbb R^F$. However, the infinite dimensional setting is still useful for waveform data, where samples have arbitrarily many entries, and can be embedded in $\ell_2$. This setting also comprises interpolated data in which individual samples are not necessarily recorded at the same interpolating points.
>
> The claim that certain properties “pass to finite truncations” is simply the argument given in App. A.5.
>
> 3) We do report comparisons to simple and strong baseline methods: SVM is chosen as a baseline method in all our experiments. Indeed, on all datasets except the remote sensing one, we test the performance against five benchmark learners - linear SVM, SVM with RBF, LogitBoost, RUSBoost, and BAGging), with a direct comparison against the method achieving the highest AUC/accuracy).
>
> Further, while there is pre-processing involved in all datasets reported on, the pre-processing includes standardization and imputation for the incomplete datasets. These pre-processing methods are standard practices in machine learning. Each dataset follows the fairly standard pipeline of: imputation (if incomplete) > normalization (except for the remote sensing data) > balancing (we also try not balancing) > training > validation
>
> 4) “State-of-the-art” describes the benchmark learners, not our method. While its formalism is generic, FINDER has no utility on conventional datasets with an established history of baselines and bespoke methods, like those in the ImageNET or CNN archives. Its entire utility is geared towards datasets that are too sparse or noisy for conventional methods. The datasets we focus on don’t necessarily possess a well-established baseline, but are nevertheless important to their respective fields, and we demonstrate that FINDER elevates SVM performance on such datasets.
>
> 5) Mathematically, the formalism applies to all discrete classification problems, since the entire point of featuring properly implies that different classes would occupy different regions in $\mathcal H$, implying spectral profiles that may be easily distinguished. However, on sparse datasets, the data is barely enough to reliably identify two classes, so we focused on testing the algorithm in that still significant domain (while claiming no major computational advances beyond that).
>
> Further, a complete eigendecomposition of the covariance operator is not necessary. It suffices to compute a compact SVD of the data matrix, and use Tausch-White’s method (as in MLS) to extend the left singular vectors to a basis of $\mathcal H_M$. Further, claimed advantages of FINDER are specifically on noisy datasets with small sample size. We emphasize that datasets with abundant samples are not within the purview of FINDER, (unless noise comes from a different source).
>
>
> QUESTIONS:
>
> 1. kernel-PCA usually requires an understanding of the geometry exhibited by the data, which is difficult even for moderately high dimensional datasets. FINDER essentially reduces the scope of feature maps to the orthogonal projections, using standard tensor product theory to make a near-optimal choice for classification.
>
> 2. The technical novelty w.r.t. mathematics is indeed quite mild and we agree that the work should be focused away from it. We do believe that some results did require statements in their most general forms, which is why we made them.
>
> 3. We are conducting a suite of numerical experiments on this question that should answer each such comment raised by the reviewers.
>
> 4. Agreed
>
> 5. Agreed

---

### Author Response · Authors · 2025-11-24
**General Comments**

We begin with thanks to the reviewers for their insightful comments and good faith criticisms of the work - all authors agreed that the reviews were of high quality, despite our disappointment at the low scores.

Reviewers seemed to be in agreement on the strengths and weaknesses of the work, so we make some overarching comments here and respond directly to each reviewer in separate comments below. All currently addressable comments are responded to in these direct rebuttals, while the remainder will be clarified by the updated manuscript uploaded after obtaining additional numerical results.

We had two major aims for FINDER. The first was to present our formal perspective on classification as a mathematical construct (even though we present no new algorithms for multi-class classifications). The second was to leverage certain straightforward mathematical results from within this construct to build algorithms that could tackle datasets of significant importance and intractability - what we called "noisy" datasets. By nature, all empirical datasets are "noisy" - we specifically chose ones where the effect of noise was dominant, such as ones with small sample sizes (biomedical ones) or the data itself being of low quality (cloud cover impacting remote sensing).

Having analyzed the reviews, it is clear that the high level setup and formalism was appreciated, while the reception to the mathematical details was universally negative w.r.t. both the novelty/significance and the misplaced relevance we seemed to be placing on them. We are now in agreement with the referees that the mathematical details were focused on in excess relative to their direct importance - many are being moved to the appendix to make space for facets that were judged to be of stronger significance. We do however claim that those details are relevant in some aspects and make the precise rebuttals on those fronts in the responses below.

We are also in agreement that the algorithms require more discussion and details in the main body itself instead of the appendix, which becomes quite doable with the change in focus vis a vis the mathematics - the updated version will reflect those changes.

The numerical experiments were judged to be presented in a haphazard manner, which was a bit disappointing and surprising to us since we claim significant ground was covered in testing all advantages and disadvantages of FINDER. Indeed, in many comments, we note significant discrepancy between what was intended and how the numerical experiments were received. We are updating some of the numerical experiments as per the proposals made by the reviewers, while the aspects we disagree on are directly rebutted upon in the specific comments below.

We end by thanking the reviewers yet again on their extensive and well considered comments and hope that our responses below alleviate a major portion of those concerns. The updated manuscript will cover the rest, especially the comments relating to the numerical experiments.

---

### Author Response · Authors · 2025-12-03
**Withdrawing the paper from consideration**

In light of the Openreview leak incident and the subsequent decisions taken by ICLR regarding the rebuttal phase, author-referee interactions, etc, we have decided to withdraw our work from consideration.

We would like to acknowledge again that the reviews seemed to be made in good faith and were of reasonable quality - and we state this despite our work being received poorly and us disagreeing with the scores allotted to our work.

We would also like to acknowledge that ICLR has been put in a tough situation and there was perhaps no good choice left (and that the decisions they have taken may very well turn out to be optimal when considering the entire situation).

However, the conditions under which we submitted our work and put in efforts for during the rebuttal phase have changed materially and significantly. The opportunity for dialogue and critical discussion has essentially vanished from the process. Further, in our opinion, the possibility that our rebuttals/arguments could meaningfully alter how our work is received at ICLR is now 0, irrespective of how well crafted and compelling those arguments might be. We waited as close to the deadline as possible to see if there would be a change to the new policy (or at least some form of meaningful engagement with our work). However, since that has not turned out to be the case, we have decided to withdraw our work from consideration.

---

### Note · Authors · 2025-12-03

**Comment:**

In light of the Openreview leak incident and the subsequent decisions taken by ICLR regarding the rebuttal phase, author-referee interactions, etc, we have decided to withdraw our work from consideration.

We would like to acknowledge again that the reviews seemed to be made in good faith and were of reasonable quality - and we state this despite our work being received poorly and us disagreeing with the scores allotted to our work.

We would also like to acknowledge that ICLR has been put in a tough situation and there was perhaps no good choice left (and that the decisions they have taken may very well turn out to be optimal when considering the entire situation).

However, the conditions under which we submitted our work and put in efforts for during the rebuttal phase have changed materially and significantly. The opportunity for dialogue and critical discussion has essentially vanished from the process. Further, in our opinion, the possibility that our rebuttals/arguments could meaningfully alter how our work is received at ICLR is now 0, irrespective of how well crafted and compelling those arguments might be. We waited as close to the deadline as possible to see if there would be a change to the new policy (or at least some form of meaningful engagement with our work). However, since that has not turned out to be the case, we have decided to withdraw our work from consideration.

**Withdrawal Confirmation:**

I have read and agree with the venue's withdrawal policy on behalf of myself and my co-authors.